# Cross-species insemination reveals mouse sperm ability to enter and cross the fish micropyle

Suma Garibova[1†], Eva Stickler[2†], Fatima AlAli[1], Maha A Abdulla[1], Abbirami Sathappan[1], Sahar I Da'as[1], Lillian Ghanem[2], Mohamed Nadhir Djekidel[1], Rick Portman[2], Matteo Avella[1,2,3]*

[1]Sidra Medicine, Research Branch, Reproductive and Perinatal Health Division, Doha, Qatar; [2]Department of Biological Science, University of Tulsa, Tulsa, United States; [3]Department of Biomedical Sciences, Qatar University, Doha, Qatar

## eLife Assessment

This **important** study reports the conservation of sperm-egg envelope binding by demonstrating successful recognition of the micropyle in fish eggs by mouse sperm. The evidence supporting the conclusions drawn is **convincing**. This study will be of interest to reproductive biologists and clinicians studying the biology of fertilization and fertility.

*For correspondence:
mavella@sidra.org

†These authors contributed equally to this work

Competing interest: The authors declare that no competing interests exist.

**Abstract** Extracellular matrices surrounding eggs in fish (chorion) and mammals (zona pellucida [ZP]) regulate gamete recognition before fertilization, though their mechanisms differ. Mouse sperm bind and cross the ZP at any site, while fish sperm cross the chorion through a funnel-shaped opening, the micropyle. To explore these divergent processes, we established cross-species insemination assays, mixing zebrafish eggs with mouse sperm. While mouse sperm could not bind to the chorion, a subpopulation successfully located and crossed the fish micropyle. Confocal and electron microscopy revealed that sperm entered the micropyle and accumulated in the zebrafish inter-chorion space. However, transgenic mouse sperm with mCherry-labeled acrosomes failed to undergo acrosome exocytosis efficiently in the micropyle, with both acrosome-intact and reacted sperm found in the inter-chorion space. Sperm entry and crossing were dependent on hyperactive motility, as sperm from *CatSperd^{Null}* mice, which fail to undergo hyperactivation, did not interact with or cross the micropyle. These findings suggest a conserved mechanism for sperm entry into the micropyle, providing a novel platform to investigate cross-species gamete interactions and uncover novel steps in fertilization.

## Introduction

Studies of interspecific gamete interaction have been historically important for elucidating the mechanisms regulating fertilization (*Vacquier, 1979*; *Vieira and Miller, 2006*; *Yanagimachi et al., 1976*), which is required for the successful development of fish and mammals (*Bhakta et al., 2019*; *Deneke and Pauli, 2021*). Despite decades of investigation, however, the understanding of the molecular basis of sperm-egg interaction remains incomplete.

In mice, upon mating, hundreds to thousands of sperm leave the uterus and cross the uterotubal junctions, which separate the uterus from the oviduct (*Avella et al., 2016*; *Ded et al., 2020*; *Suarez, 2016*). These sperm bind to the epithelial cells of the isthmus in the oviduct until eggs are released from the ovary during ovulation into the ampulla of the oviduct (*Hunter, 1993*), where fertilization

**eLife digest** Conception begins when a sperm cell fuses with an egg cell in a process called fertilization, initiating the development of a genetically unique embryo. This fundamental event is conserved among vertebrates, although the cellular environments and molecular mechanisms involved vary across taxa, especially between mammals and externally fertilizing fish.

In mammals, sperm must first bind to and penetrate the zona pellucida, a matrix surrounding the egg. They then attach to the membrane underneath, called the oolemma, and initiate fusion. In teleost fish, by contrast, the egg is surrounded by a structurally distinct extracellular envelope called the chorion. Unlike mammalian sperm, fish sperm do not bind directly to this barrier. Instead, they gain access to the egg through a narrow, funnel-shaped canal called the micropyle.

Until recently, it was widely believed that the mechanisms that help the sperm recognize and respond to eggs were specific to different species and incompatible across distant taxa. However, Garibova et al. provided evidence that some of the signalling cues guiding sperm towards the eggs may be conserved across vertebrates.

The researchers investigated whether mouse sperm can recognize and respond to the structural features of zebrafish eggs. To test whether mouse sperm could locate and enter the micropyle, they used time-lapse imaging, as well as confocal and electron microscopy.

The results revealed that while mouse sperm did not bind to the fish chorion, they could locate and enter the zebrafish micropyle. Moreover, a subset of mouse sperm actively swam toward and across the micropyle, a behavior analogous to that of zebrafish sperm, which become active only in proximity to egg cells.

Further molecular analysis revealed that sperm cells needed to be hyperactive to complete their passage through the chorion. This hyperactivity was regulated by a calcium channel called CatSper. Using sperm from mutant mice lacking CatSper, the cells were unable to reach or cross the micropyle. These findings suggest that despite differences in egg coats between mammals and fish, the ways sperm find and navigate toward the egg may rely on evolutionarily conserved mechanisms.

Such discoveries could aid scientists and clinicians working in assisted reproductive technologies in humans, as well as in companion and farm animals, by providing insights into the molecular mechanisms underlying sperm guidance and egg recognition. To enable clinical or agricultural translation of these findings, future studies should identify the molecular mediators governing sperm guidance and micropyle recognition. Functional validation through loss-of-function assays in transgenic fish and mouse models will be crucial to determine whether these mechanisms are truly conserved across taxa. A deeper understanding could inform the development of novel sperm selection strategies or contraceptive approaches aimed at modulating gamete interaction.

---

occurs. A portion of these sperm undergoes hyperactivation, which consists of a sudden change in the motility pattern of the sperm flagella, generating an asymmetrical beat that mediates the sperm release from the isthmus (*Han et al., 2016*). Hyperactive sperm enter the ampulla of the oviduct and traverse the cumulus mass, a hyaluronic-interspersed mass of cells, remnant of the granulosa cells from the follicle. Each cumulus mass in the ampulla may enclose a single Metaphase II (MII) egg, mature and competent for fertilization. Upon reaching the egg, sperm bind to the zona pellucida, an extracellular glycoprotein matrix composed of three glycoproteins defined as ZP1, ZP2, and ZP3. After binding, the fertilizing sperm cross the zona and fuse with the oolemma (*Bhakta et al., 2019*).

In fish, the sperm do not bind to the chorion surrounding the ovulated egg (*Mold et al., 2001*). Instead, sperm cross the chorion through a narrow, funnel-shaped canal defined as the micropyle (*Yi et al., 2019*). Early studies in the Pacific herring (*Clupea* sp.) show that, once released into seawater, sperm are virtually motionless until they come near the micropyle area of an egg, whereupon they become motile (*Yanagimachi, 1957*; *Yanagimachi et al., 2017*). In bitterling, zebrafish, and other fish species, sperm contact with water is sufficient to activate their motility, yet, as they pass near the micropyle area, their motility increases (*Suzuki, 1958*).

Because fish sperm bypass binding to the chorion using the micropyle, fish zona proteins have always been assumed to lack the ability to support sperm binding (*Mold et al., 2001*). Zebrafish chorion is composed of two proteins, ZP2 and ZP3, homologs to the mammalian zona proteins (*Mold*

*et al., 2001*). Like other external fertilizing species, zebrafish sperm cross the chorion through the micropyle before fusing with the egg. On the other hand, mammalian eggs do not present a micropyle in the zona pellucida. Hence, sperm can cross at any binding site of the zona matrix, as shown by the presence of vesiculated acrosomal shrouds in multiple sites of the zona surface (*VandeVoort et al., 1997*; *Wakayama et al., 1996*; *Yanagimachi and Phillips, 1984*).

The abovementioned observations raise the prediction that mouse sperm may not recognize zebrafish ZP2 or ZP3 and would therefore be unable to bind to the zebrafish chorion. Additionally, mouse sperm may not recognize the micropyle, given that no such structure exists in mammalian zonae pellucidae. Importantly, these two predictions have never been experimentally tested. As testing these predictions could provide valuable insights into the molecular basis of species specificity in gamete attraction and recognition prior to fertilization, the overall aim of this study was to determine whether mammalian sperm can recognize and respond to zona proteins or the micropyle in zebrafish eggs.

To test these predictions, we have established a mouse-fish gamete insemination assay that consists of mixing fish eggs with mouse sperm. We show that mouse sperm could not bind to individual zebrafish ZP2/ZP3 nor to the chorion. Yet, unexpectedly, a subpopulation of mouse sperm recognized and entered the micropyle and accumulated in the oocyte inter-chorion space (ICS; the space between the inner aspects of the chorion and the oocyte). To capitalize on such discoveries, we used mouse genetics to show that only hyperactive mouse sperm could enter and cross the micropyle. We also documented that the zebrafish micropyle could not effectively induce acrosome exocytosis. Our studies provide a novel platform for cross-species insemination experiments and have implications for the mechanisms by which sperm can locate and enter the micropyle of the egg.

## Results
### Zebrafish zona pellucida proteins are not sufficient to support mammalian sperm binding

Mouse zona pellucida is composed of three glycoproteins: ZP1, ZP2, and ZP3, and the N-terminus of ZP2 has been identified as the ligand necessary and sufficient to support mouse sperm binding (*Avella et al., 2014*; *Tokuhiro and Dean, 2018*). In contrast, the zebrafish chorion contains only two glycoproteins, ZP2 and ZP3 (*Mold et al., 2001*), and zebrafish sperm do not rely on these proteins for interaction with the chorion. Therefore, while zebrafish ZP2 and ZP3, either individually or in combination, are not predicted to mediate sperm interaction, this has not been experimentally tested and remains unknown. To assess this, we used mouse sperm, which are known to require the N-terminus of ZP2 for binding to the zona pellucida (*Avella et al., 2014*; *Tokuhiro and Dean, 2018*), as a functional assay to test the ability of zebrafish ZP2 and ZP3 to support sperm binding.

Zebrafish ZP2 isoforms present 31.09–33.09% identity to mouse ZP2 (*Figure 1A*). The first 6-cysteine residues of zebrafish ZP2 are organized into a trefoil domain, and zebrafish ZP2 lacks the homologous domain to the mammalian N-terminal region (*Figure 1A*), which has been reported to be necessary and sufficient to support mouse and human sperm binding (*Avella et al., 2014*; *Tokuhiro and Dean, 2018*). Recombinant baculovirus encoding zebrafish ZP2 and ZP3 was generated (*Figure 1B*). Each recombinant protein presented at the N-terminus a gp67 signal peptide from *Autographa californica* nuclear polyhedrosis virus (AcMNPV) (38 aa) to ensure proper secretion, and at the C-terminus, a 6-His tag to enable purification. Zebrafish ZP2$^{25-405}$ and ZP3$^{22-396}$ start at the N-terminus of the secreted ectodomains and include an even number of cysteine residues (*Figure 1A*). Confocal images and quantification with boxplots showed that mouse sperm bound sparsely to beads coated with either zebrafish ZP2 or ZP3 and to beads coated with both zebrafish ZP2 and ZP3, yet bound efficiently to beads coated with mouse ZP2 (*Figure 1C and D*; *Avella et al., 2016*). Therefore, we could conclude that beads coated with individual zebrafish ZP2 or ZP3, or with both ZP2 and ZP3, cannot support sperm binding in vitro as efficiently as mouse ZP2 (*Figure 1D*).

### Mouse sperm do not bind to the zebrafish chorion
To further validate these observations, we used native zebrafish chorion (*Mold et al., 2001*). First, to test whether zebrafish chorion remains intact under mouse in vitro fertilization (IVF) conditions, zebrafish ovulated eggs (n=100, 3 replicates) were incubated at 37°C, 5% $CO_2$ in HTF/HSA (mouse

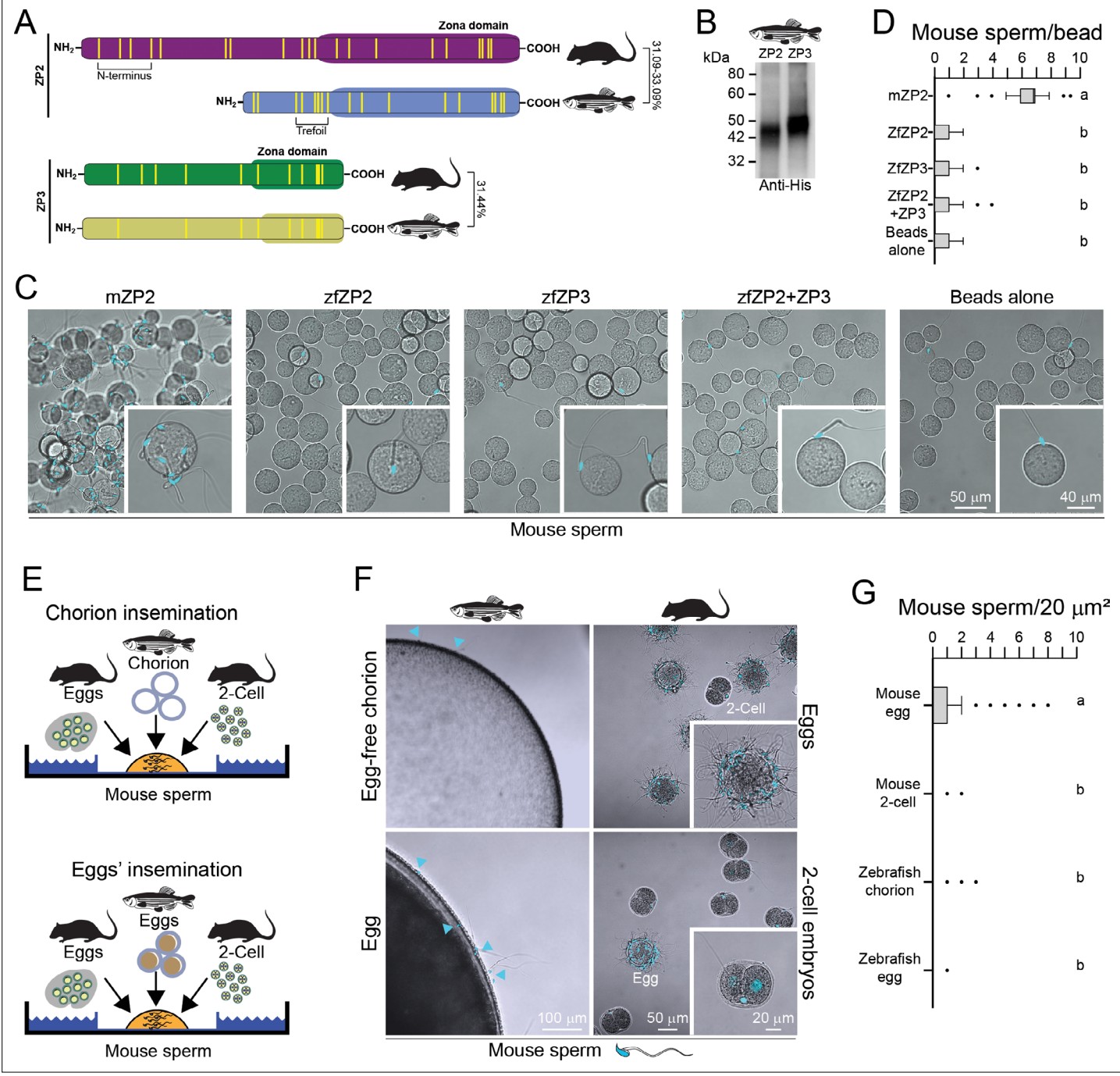

**Figure 1.** Mouse sperm do not bind to zebrafish zona proteins. (**A**) Schematic of zebrafish ZP2 and ZP3 peptides compared to mammalian homologs (Clustal Ω). Yellow bars represent cysteine residues. Percent values indicate identity between mouse and zebrafish amino acid sequences (NP_035905.1, mouse ZP2; AAK16578.1, zebrafish ZP2, variant A; AAK16577.1, zebrafish ZP2, variant B; AAK16579.1, zebrafish ZP2, variant C; NP_035906.1, mouse ZP3; NP_571406, zebrafish ZP3). (**B**) SDS-PAGE of recombinant zebrafish ZP2, ZP3, or ZP2 and ZP3 peptides (6-His mAb, immunoblot) expressed in Sf9 cells after purification from agarose beads. Molecular mass is indicated on the left. (**C**) Capacitated mouse sperm binding to beads carrying zebrafish (zf) ZP2, zfZP3, or zebrafish zfZP2 and zfZP3. Beads carrying mouse (**m**) ZP2 N-termini and beads alone were used as positive and negative controls, respectively. Differential interference contrast (DIC) (top) and confocal z projection (bottom) images, sperm nuclei (blue) stained with Hoechst. (**D**) Boxplots represent the median (vertical line) number of mouse sperm binding to mammalian/fish peptide beads or beads alone and data points within the 10th and 90th percentiles (error bars). Boxes include the middle two quartiles, and dots indicate the outliers. Superscript letters show statistical significance (p<0.05) defined by one-way ANOVA followed by Tukey's HSD (honestly significant difference) post hoc test. (**E**) Schematic of cross-species insemination assays using mouse sperm to inseminate zebrafish chorion (top) or eggs (bottom). (**F**) Representative pictures of zebrafish chorion or eggs inseminated with mouse sperm. Top-right panel: mouse sperm binding to normal mouse eggs after 60 min of incubation. Inset, ×2.0 magnification. Mouse two-cell

*Figure 1 continued on next page*

**eLife** Research article

Cell Biology | Developmental Biology

*Figure 1 continued*

embryos serve as a negative control for sperm binding. Bottom-right panel: mouse sperm binding to mouse two-cell embryos; mouse eggs serve as an internal positive control for mouse sperm binding. Top-left panel: mouse sperm incubated with zebrafish chorion without ovulated oocyte (60 min incubation). Bottom-left panel: mouse sperm incubated with zebrafish ovulated oocytes (60 min incubation). (**G**) Boxplots represent the median (vertical line) number of mouse sperm binding to mammalian/fish oocytes or embryos (sperm bound per 20 μm² projected surface area) and data points within the 10th and 90th percentiles (error bars). Boxes include the middle two quartiles, and dots indicate the outliers. Superscript letters show statistical significance (p<0.05) defined by one-way ANOVA followed by Tukey's HSD post hoc test. D and G, three independent biological replicates.

The online version of this article includes the following source data and figure supplement(s) for figure 1:

**Source data 1.** Original uncropped, unedited blot file.

**Source data 2.** Uncropped blot with molecular markers and zebrafish ZP2 and ZP3 protein bands labeled.

**Figure supplement 1.** Zebrafish eggs in human tubal fluid/human serum albumin (HTF/HSA).

**Figure supplement 2.** Cross-species insemination of ovulated mouse Metaphase II (MII) oocytes with zebrafish sperm.

IVF conditions) and observed over time (0, 30, 60, 120, and 240 min) for morphological signs of deterioration of the chorion. Control eggs (n=30, 3 replicates) were maintained in Hank's solution (*Westerfield, 2007*) at room temperature (*Figure 1—figure supplement 1*). Under mouse IVF conditions, zebrafish oocytes presented no signs of zona deterioration within the first 60 min (*Figure 1—figure supplement 1*). Therefore, we used 60 min as the time limit for incubating zebrafish chorion or eggs with mouse sperm (previously incubated for 45 min at 37°C, 5% $CO_2$ in HTF/HSA).

Zebrafish ovulated eggs were collected and preserved in Hank's saline solution to prevent egg activation (*Westerfield, 2007*). As HTF contains water, it may induce activation of zebrafish oocytes, which may result in postfertilization biochemical modification of the ZP proteins of the chorion (*Masuda et al., 1991*). Thus, we mechanically isolated the chorion from zebrafish eggs in Hank's, moved them to HTF/HSA, and inseminated them with $10^5$ progressive motile mouse sperm (*Figure 1E*). The isolated chorion was found to not support mouse sperm binding (*Figure 1F*, top-left). To quantify these observations that involve gametes of considerably different sizes (~80 μm for mouse egg/two-cell embryo vs. ~700 μm for zebrafish egg), we acquired z maximum intensity projections of inseminated eggs/embryos by confocal microscopy (LSM800, Zeiss), and calculated the number of sperm bound per 20 μm² area using the ZEN 3.2 software (*Figure 1G*). Our quantitative analyses using boxplots showed that mouse sperm bind to the fish chorion (0.025±0.11, s.e.m.) with significantly lower efficiency to mouse zonae (0.76±0.05, s.e.m.) (*Figure 1F*, top-right), and with comparable efficiency to the zonae surrounding mouse two-cell embryos (0.041±0.01, s.e.m.) (*Figure 1F*, bottom-right).

In addition, whole ovulated zebrafish oocytes surrounded by a chorion were placed in HTF/HSA and inseminated with $10^5$ progressive motile mouse sperm (37°C, 5% $CO_2$ in HTF/HSA) for 60 min. Inseminated zebrafish oocytes showed very few sperm bound to the chorion (*Figure 1F*, bottom-left). Quantification of these observations found that mouse sperm bind to the chorion surrounding zebrafish eggs with comparable efficiency (0.06±0.02 s.e.m.) as to the chorion only (no oocyte within) isolated from zebrafish eggs in Hank's solution (*Figure 1G*). From these observations, we conclude that mouse sperm cannot bind to the zebrafish chorion.

## Zebrafish sperm cannot bind mouse zonae pellucidae

To evaluate whether zebrafish sperm can recognize and bind to the mammalian zona pellucida, we performed cross-species insemination experiments using ovulated, cumulus-free mouse oocytes incubated in water with zebrafish sperm. Ovulated mouse MII oocytes (n=10; 3 replicates) were denuded of cumulus cells using hyaluronidase, then gently acclimated from standard mouse IVF conditions (HTF/HSA) to Hank's solution and subsequently to water, to mimic the zebrafish insemination environment. Then, mouse eggs were incubated with zebrafish sperm using fish IVF conditions as described in the Materials and methods section. No zebrafish sperm bound to the mouse zona pellucida (*Figure 1—figure supplement 2*). These results support the hypothesis that zebrafish sperm do not interact with mouse zonae.

## Mouse sperm recognize the micropylar region of fish oocytes

Although the chorion does not support mouse sperm binding, the micropyle region of ovulated oocytes is known to attract fish sperm (*Yanagimachi, 1957*; *Yanagimachi et al., 2013*). This is due to

the presence of still unidentified glycosylated protein(s) present in the micropylar region of fish eggs (*Figure 2A and B*).

Previous studies have yielded conflicting results regarding the presence of glycoproteins within the zebrafish micropyle. *Yanagimachi et al., 2017*, reported the absence of detectable glycoproteins in the micropyle, whereas *Dingare et al., 2018*, identified glycoprotein expression localized to the micropyle region. Given the critical role of the micropyle in directing sperm entry during external fertilization in teleosts, the presence or absence of a specific glycoprotein could have important functional implications in gamete attraction and fertilization. To address this discrepancy, we investigated whether micropyle protein (MP) is indeed present at the zebrafish micropyle and evaluate the potential role of the MP in guiding or facilitating sperm entry.

To test for the presence of the MP in zebrafish, we stained freshly ovulated oocytes with fluorescently conjugated wheat germ agglutinin (WGA-633). We confirmed the presence of a positive signal in the micropylar region of zebrafish oocytes (*Figure 2A*), as previously described (*Dingare et al., 2018*). Using fish IVF conditions, we inseminated zebrafish ovulated eggs with zebrafish sperm and imaged sperm around the micropylar area and within the inner opening of the micropyle (*Figure 2B*). The presence of sperm after insemination was found to depend on the presence of the MP. Indeed, when the MP was biochemically removed with trypsin (*Figure 2C and D*), the number of zebrafish sperm surrounding the micropylar region or entering the micropyle significantly decreased (*Figure 2E and F*), which also led to a reduction in the fertilization rates (*Figure 2G*).

While performing our cross-species insemination assay, upon in vitro insemination of zebrafish ovulated oocytes with mouse sperm (37°C, 5% $CO_2$, HTF/HSA), we could observe the accumulation of mouse sperm to the chorion area surrounding the micropyle (*Figure 3A*, left panel). This binding persisted after extensive washes with a glass pipette, which was able to detach loosely bound mouse sperm to mouse two-cell embryos (negative control; *Figure 1F*).

Then, to assess the ability of the zebrafish micropyle to support mouse sperm binding independently from the oocyte, we mechanically isolated the chorion in Hank's solution to prevent oocyte activation. The micropylar region of the isolated chorion supported mouse sperm binding (*Figure 3A*, right panel). The binding pattern of the mouse sperm was comparable with zebrafish sperm binding around the micropyle region (*Figure 2B*). Upon quantification of these observations with boxplots, we found that mouse sperm bound the micropylar region (1.11±0.1, s.e.m.) with comparable efficiency to mouse zonae pellucidae (1.07±0.07, s.e.m.) surrounding ovulated eggs (*Figure 3B*), while sperm bound to the chorion away from the micropyle (at a radial distance greater than 160 μm from the opening of the micropyle) were rarely observed (0.2±0.04, s.e.m.). Trypsin digestion of the chorion virtually eliminated the MP as shown by the absence of WGA-633 staining (*Figure 2C*). When inseminated with mouse sperm, trypsin-treated micropyle regions did not support sperm binding (*Figure 3—figure supplement 1A and B*). From these observations, we concluded that mouse sperm can recognize the fish micropyle in vitro, and such recognition is dependent on the presence of the MP.

It is conceivable that, if the MP exerts a chemotactic effect on zebrafish and mouse sperm, it would be expected that this factor produces a concentration gradient. To measure if a fluorescence gradient may exist from the periphery toward the micropyle inner opening, via ImageJ/Fiji, we measured the raw integrated density (RID) (*Brazill et al., 2018*) of the zebrafish WGA-633-stained chorion encompassing the micropyle (n=10) (*Figure 3C*). We found that fluorescence density increases from the periphery toward the central 40 μm area encompassing the micropyle region (*Figure 3D*, left-Y axis). Of note, we observed an increasing number of mouse sperm bound to the chorion from the periphery toward the micropyle (*Figure 3D*, right-Y axis). The direct correlation between the increasing fluorescence density and the number of sperm bound from the periphery toward the micropyle is consistent with the potential role of the MP as a chemoattractant to guide sperm toward the micropyle (*Yanagimachi, 1957*; *Yanagimachi et al., 2013*).

To capture live mouse sperm entering the micropyle, we inseminated 20 zebrafish ovulated eggs in 300 μl HTF (Merck Millipore, USA) with 3000 progressive motile sperm and performed four different independent experiments using mouse sperm from three different fertile males.

The recordings spanned a total period of 63 min and 17 s, with each video lasting an average of 16±2 min. In total, 546 sperm were observed across the four videos. The results showed that only a subpopulation of sperm was able to enter the micropyle canal (8.25±3.63, 5.92%±2.38%), while the majority were found not entering the micropyle and either swimming close to the micropyle without

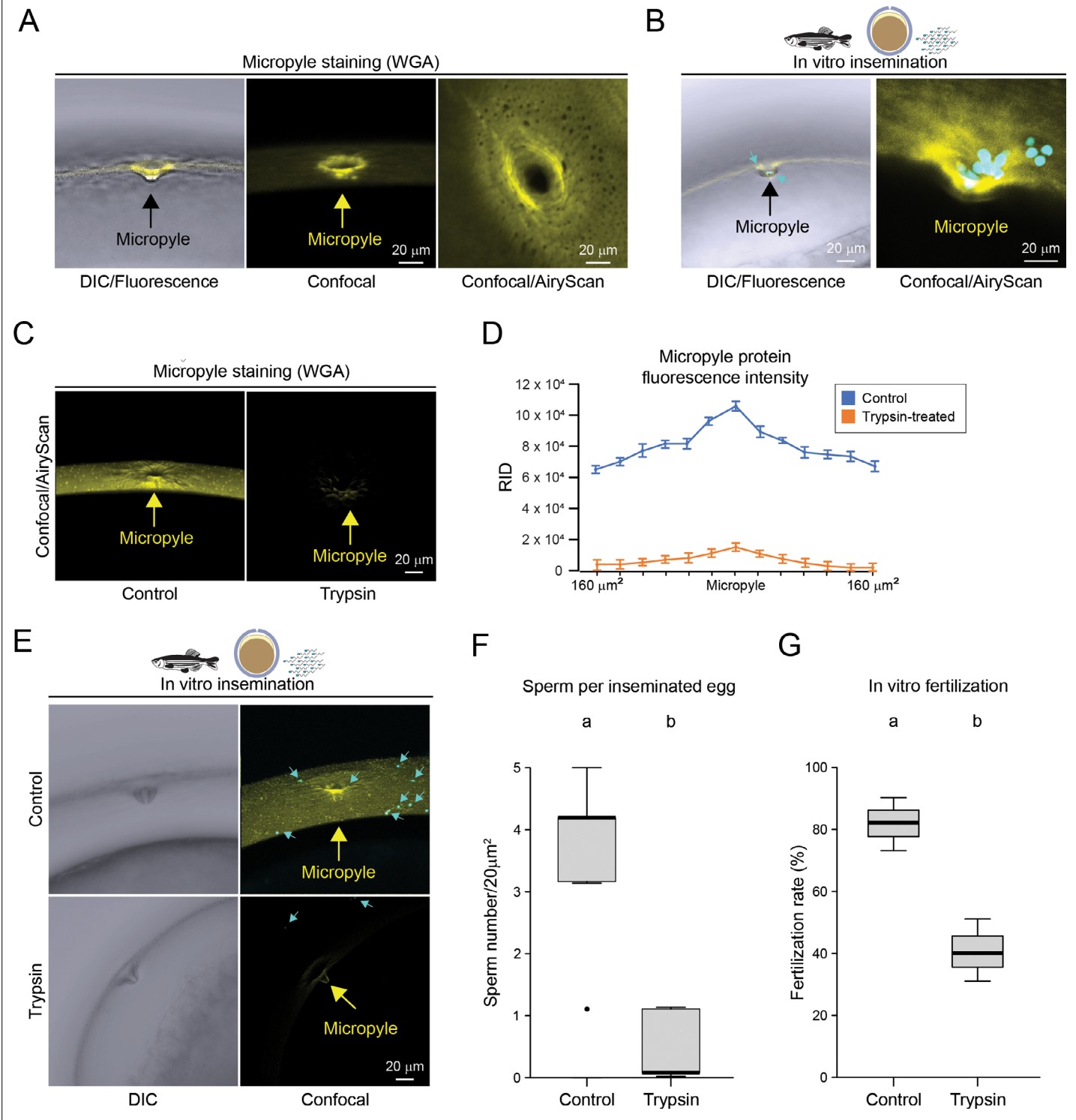

**Figure 2.** Fish sperm interaction with the micropyle. (**A**) Confocal images of the micropyle stained by WGA-633 in zebrafish oocytes (n>15); arrows indicate the micropyle. (**B**) Zebrafish in vitro insemination: Hoechst-stained zebrafish sperm (light blue) that have approached or entered the micropyle in freshly ovulated oocytes (yellow, WGA-633-stained); samples were fixed in paraformaldehyde few seconds after insemination. (**C**) Zebrafish eggs, untreated (left), or treated with trypsin to eliminate the micropyle protein (right). (**D**) Fluorescence was measured (Fiji/ImageJ) within a 20 µm² area; 0 indicates the micropyle opening position (yellow), 160 µm indicates the most distant position measured from micropyle opening. (**E**) Same as in (**C**); left, DIC, right, confocal images (maximum intensity projection) of zebrafish oocytes inseminated with zebrafish sperm. (**F**) Same as in *Figure 1G*, for the quantification of the number of zebrafish sperm approaching and entering the micropyle of oocytes treated/not treated (control) with trypsin (n=3). (**G**) Same as in *Figure 1G*, for fertilization rates (n=3).

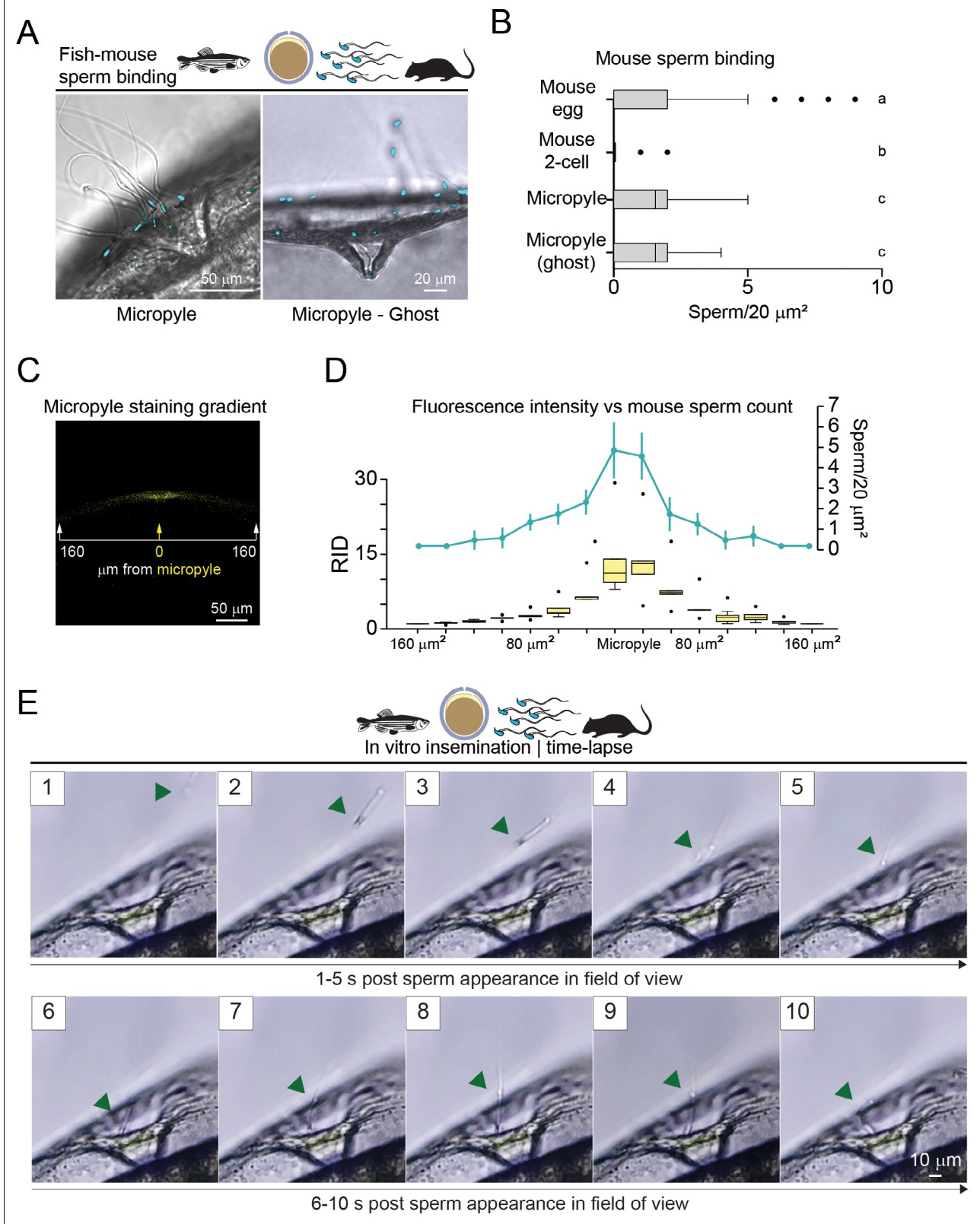

**Figure 3.** Mouse sperm cross the fish micropyle. (**A**) Cross-species insemination: mouse sperm (Hoechst-stained, light blue) in the zebrafish micropyle region of a chorion (60 min incubation) surrounding the oocyte (left) or of a chorion mechanically freed from the oocyte (right, Ghost). (**B**) Quantification of mouse sperm in the micropyle region of chorion with or without zebrafish oocyte: same as in *Figure 1G*; mouse eggs or two-cell embryos served as an internal positive and negative control for sperm binding (n=3). (**C**) X-Y plane confocal projection of zebrafish chorion encompassing the WGA-

*Figure 3 continued on next page*

*Figure 3 continued*

633 (yellow) micropyle region (~320 µm); bar with arrows indicates positions at which fluorescence was measured (Fiji/ImageJ) as in *Figure 2D*. (**D**) Quantification of mouse sperm across the micropyle region: boxplots represent the median (vertical line) raw integrated density ratio (RID ratio, left Y axis) measured on 10 zebrafish chorions; data points within the 10th and 90th percentiles (error bars). Boxes include the middle two quartiles, and dots indicate the outliers. A light blue line represents the number of sperm (right Y axis) found in the corresponding chorion position (X axis); error bars represent s.e.m.; statistical significance (p<0.05) across RID ratios or sperm numbers in different positions is defined by one-way ANOVA followed by Tukey's HSD (honestly significant difference) post hoc test. (**E**) Time-lapse frames from *Figure 3—video 1*, showing the first sperm entering the micropyle of a freshly ovulated zebrafish egg. Green arrowheads indicate mouse sperm. Insets show the number of seconds ('s') after the first sperm appears in the field of view.

The online version of this article includes the following video and figure supplement(s) for figure 3:

**Figure supplement 1.** Trypsin treatment of zebrafish micropyle.

**Figure supplement 2.** Mouse sperm bypassing the oocyte, swimming close without interacting with the micropyle or entering the micropyle canal.

**Figure supplement 3.** Time-lapse confocal imaging of micropyle detachment under mouse in vitro fertilization (IVF) conditions.

**Figure 3—video 1.** The entry of mouse sperm into the micropyle of an ovulated zebrafish egg.

https://elifesciences.org/articles/106303/figures#fig3video1

**Figure 3—video 2.** Simultaneous entry of two sperm into the micropyle canal of a zebrafish egg.

https://elifesciences.org/articles/106303/figures#fig3video2

entering (110.5±44.04, 79.20%±5.42%) or passing by and completely ignoring the zebrafish oocyte (17.75±4.22, 15.00%±6.28%)(quantification performed as displayed in *Figure 3—figure supplement 1*).

Mouse sperm reached the zebrafish oocytes 48.5±10.6 s post insemination and entered the micropyle canal within 24.5±7.1 s after approaching the zebrafish oocyte, attempting to cross the micropylar opening (*Figure 3E*, *Figure 3—figure supplement 2*, and *Figure 3—videos 1–2*). Under mouse IVF conditions, the micropyle detached 6–7 min after imaging started and continued progressively over time (*Figure 3—figure supplement 3*).

## Mouse sperm cross the fish micropyle opening

In mammalian fertilization, once sperm bind to the zona pellucida, they cross it to reach the perivitelline space (PVS), a necessary step prior to fusion with the oolemma. Previous studies using cross-species insemination assays have shown that when mammalian sperm are able to bind the egg coat of another species, they can cross its extracellular matrix (*Baibakov et al., 2012*; *Yauger et al., 2011*). For instance, human sperm have been shown to cross the zona pellucida of transgenic mouse eggs expressing human ZP2 and accumulate within the PVS, as human sperm do not fuse with the mouse oolemma (*Baibakov et al., 2012*). Given our observation that mouse sperm interact with the micropyle region and are capable of entering the micropyle canal, we investigated whether they could similarly cross the micropyle and reach the ICS.

Using confocal microscopy (Zeiss LSM 800), we found that multiple mouse sperm (115.87±0.01) entered the micropyle canal and crossed the micropyle opening, accumulating in the ICS (*Figure 4A*, *Figure 4—figure supplement 1*; n=12–14 eggs, 3 replicates), representing 0.11587 ±0.0001% of the total inseminated progressively motile sperm. After crossing, sperm concentrated either within 160 µm radius around the micropyle opening (*Figure 4A*, left and mid panels) or away from the micropyle (farther than 160 µm from the micropyle opening; *Figure 4A*, right panel). Quantifications of these observations revealed a comparable, yet significantly higher, number of ICS mouse sperm around the micropyle than away from the micropyle (*Figure 4B*).

We then wondered at what site of the micropyle region mouse sperm crossed the chorion. Previous studies have shown that, after zona binding, mammalian sperm can cross the matrix at any binding site (*VandeVoort et al., 1997*; *Wakayama et al., 1996*; *Yanagimachi and Phillips, 1984*), whereas fish sperm cross the chorion only at the micropyle inner opening (*Yi et al., 2019*). Thus, we investigated whether mouse sperm cross the chorion at any site of the micropyle region or, like fish sperm, use the inner opening of the micropyle. Using scanning electron microscopy, we observed mouse sperm within the micropyle canal (*Figure 4C*, left). However, no mouse sperm were found mid-way through the chorion in the micropyle region, indicating that mouse sperm likely used the micropyle inner opening to cross the fish chorion. To validate this thesis, we used transmission electron microscopy (TEM) to

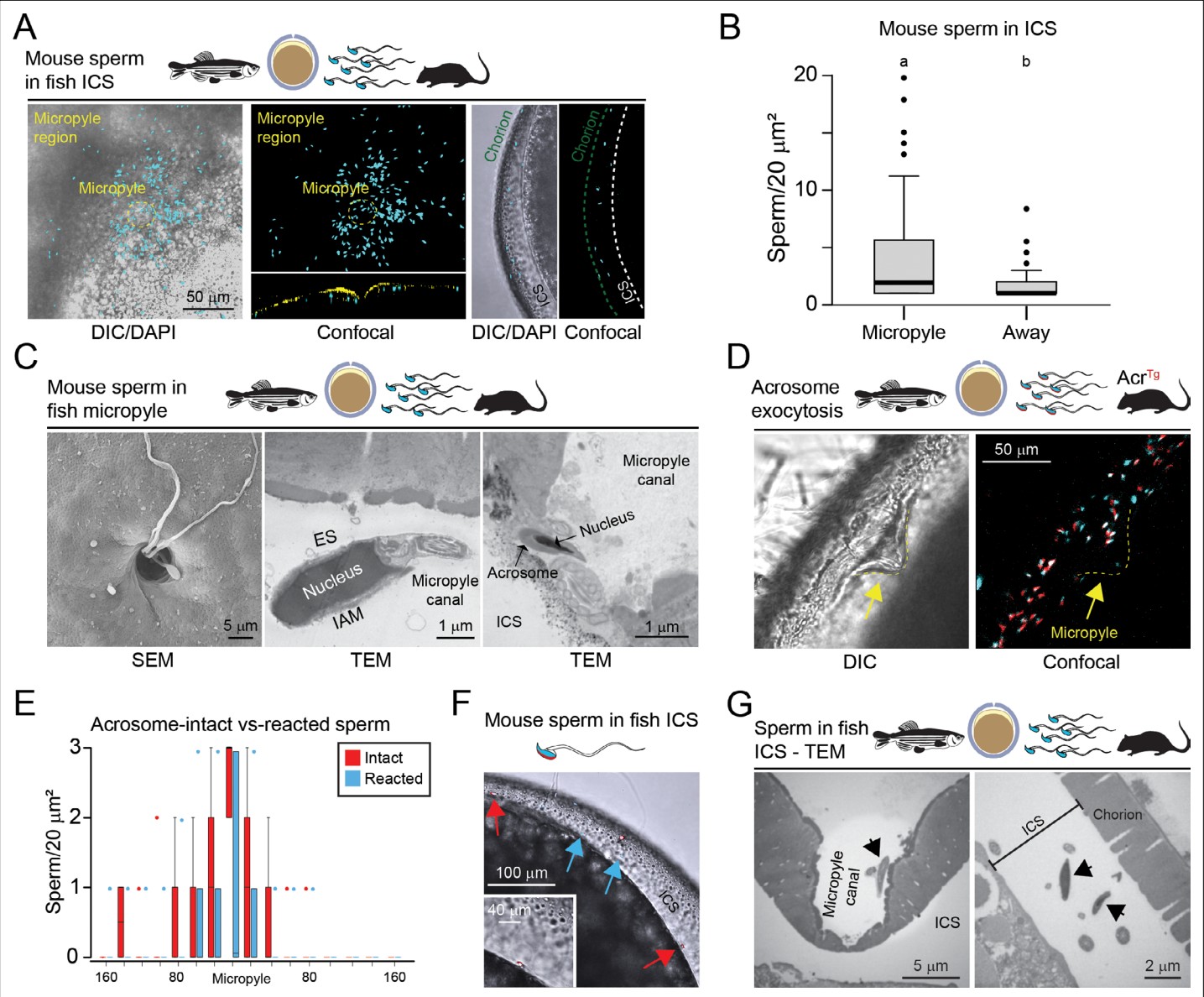

**Figure 4.** Localization and dynamics of sperm interactions with the micropyle and inter-chorion space (ICS) in zebrafish oocytes. (**A**) Hoechst-stained sperm (blue) which has crossed the micropyle. Differential interference contrast (DIC) (left) and confocal (mid-panel) projection of sperm accumulated in the ICS of a zebrafish oocyte imaged from the top; in the left and middle/top panels, the 633 channel is turned off to visualize the sperm accumulated in the ICS around the micropyle region (yellow circled); inset shows a longitudinal section of the same oocyte, showing the Hoechst-stained sperm (blue) under the WGA-633-stained micropyle region (yellow). Right, Hoechst-stained sperm (blue) is included in the ICS. (**B**) Quantification as in *Figure 1D* of mouse sperm in the ICS below (left) or away (right) from the micropyle region. Error bars represent s.e.m., statistical significance (p<0.05) is defined by one-way ANOVA followed by Tukey's HSD (honestly significant difference) (n=3). (**C**) Electron microscopy of mouse sperm in the zebrafish micropyle region and within the micropyle canal, 1 hr after insemination. Left, scanning electron microscopy (SEM), mid and right panels, transmission electron microscopy (TEM); IAM, inner acrosomal membrane; OAM, outer acrosomal membrane; ES, equatorial segment. (**D**) Acr^Tg sperm in the micropyle region. Left, DIC; right, confocal projection. The yellow arrow indicates the micropyle opening. (**E**) As in *Figure 1G*, with Acr^Tg sperm (intact vs. reacted). (**F**) Acrosome-intact (red arrows) and reacted (light-blue arrows) mouse sperm in zebrafish ICS. Inset represents a ×2 magnification of an acrosome-intact sperm in the ICS. (**G**) Left panel, micropyle structure after the entry of multiple mouse sperm. Right panel, acrosome-intact sperm from the ICS of the oocyte, the micropyle of which is shown in the left panel. Black arrows, mouse sperm.

The online version of this article includes the following video and figure supplement(s) for figure 4:

**Figure supplement 1.** Orthogonal view generated using ZEN Lite software (Zeiss, Germany), showing z-stack sections at three focal planes.

**Figure supplement 2.** Interaction of conserved mouse and fish sperm trimmers with mammalian or fish egg proteins.

**Figure 4—video 1.** Mouse sperm interaction with zebrafish egg's inter-chorion space (ICS).

*Figure 4 continued on next page*

*Figure 4 continued*

https://elifesciences.org/articles/106303/figures#fig4video1

**Figure 4—video 2.** Mouse sperm in the zebrafish inter-chorion space (ICS).

https://elifesciences.org/articles/106303/figures#fig4video2

**Figure 4—video 3.** High-resolution view of mouse sperm zebrafish inter-chorion space (ICS) area.

https://elifesciences.org/articles/106303/figures#fig4video3

study mouse sperm in the micropyle canal (*Figure 4C*, mid and right). TEM analyses confirmed that mouse sperm did not cross the chorion at the binding site. From this, we concluded that the interaction between mouse sperm and the zebrafish micropyle may differ from the mechanism of sperm binding with the zona pellucida. To that end, we used sperm from transgenic mice to further characterize the nature of this interaction.

## Ineffective induction of mouse sperm acrosome exocytosis during interaction with the fish micropyle

In mammals, acrosome exocytosis is a necessary step in fertilization. A fusion event between the plasma membrane and the outer acrosomal membrane enables the release of acrosomal contents and exposes the inner acrosomal membrane along with the equatorial segment—a remnant of the sperm plasma membrane—which is necessary for subsequent fusion with the oolemma. In contrast, sperm from most teleost species, including zebrafish, lack an acrosome entirely, and thus do not rely on this mechanism for fertilization. However, in a few chondrostean fish species such as sturgeon, acrosome-bearing sperm undergo acrosome exocytosis specifically within the micropyle canal just prior to gamete fusion (*Psenicka et al., 2010*). The absence of an acrosome in zebrafish sperm suggests that their micropyle may not possess the molecular signals required to induce this process (*Hirohashi and Yanagimachi, 2018*). While such a conclusion is plausible, it has not been directly tested. To address this, we examined whether the zebrafish micropyle can elicit acrosome exocytosis in mouse sperm, which indeed presents an acrosome. This experiment was designed to determine whether the micropyle microenvironment provides inductive cues sufficient to trigger this exocytotic event or whether acrosome exocytosis is strictly mediated by species-specific gamete interactions. To determine whether the zebrafish micropyle can induce mouse sperm acrosome exocytosis, we used TEM and confocal microscopy to track the status of the acrosomes of mouse spermatozoa during cross-species insemination. We used transgenic mouse sperm that accumulate mCherry in the acrosome (*Avella et al., 2016*) (*Acrosin^{Tg}*). Acrosome-intact sperm present a visible fluorescent acrosome beneath the nucleus (*Avella et al., 2016*). During acrosome exocytosis, mCherry is released upon fusion of the sperm plasma membrane with the outer acrosomal membrane. This results in the loss of fluorescent signal and marks these sperm as 'acrosome-reacted', as documented by fluorescent and electron microscopy analyses (*Avella et al., 2016*). Through confocal microscopy, we found 86 ± 4% (n=10, s.e.m.) of sperm surrounding the micropyle to be acrosome intact (*Figure 4D and E*). Sperm could cross the micropyle inner opening irrespective of their acrosomal status (reacted or intact; *Figure 4C*, mid and right, respectively). Additionally, both acrosome-intact and acrosome-reacted sperm were found in the ICS via confocal microscopy and TEM (*Figure 4F and G*), though they were never observed to fuse with the egg cell under our experimental conditions (*Figure 4D*), possibly due to predicted low affinity between the Izumo1-Spaca6-Tmem81 mouse sperm trimeric complex (*Deneke et al., 2024*) with the zebrafish egg cell protein Bouncer. Indeed, based on AlphaFold2-Multimer structural predictions, we observed low binding affinity between zebrafish Bouncer and the mouse trimer (Izumo1+Spaca6+Tmem81) as indicated by low interface-predicted template modeling (ipTM) scores and high predicted aligned error (PAE) values (*Figure 4—figure supplement 2*).

Thus, we concluded that mouse sperm can recognize the micropylar region of zebrafish oocytes and cross the micropyle inner opening regardless of the acrosome status, though passage through the micropyle does not appear to induce acrosome exocytosis.

## Hyperactive motility is necessary to mediate sperm crossing the zebrafish micropyle

We found that mouse sperm in the fish ICS were still motile and presenting what may appear to be a hyperactive motility pattern (*Figure 4—videos 1–3*).

In mammals, hyperactivation is necessary for crossing the zona (*Miller, 2024*). To investigate whether hyperactivated motility is also required for mouse sperm to locate or traverse the zebrafish micropyle, we performed cross-species insemination experiments using sperm from *Catsperd* $^{Null}$ mice. *CatSperd* encodes a pore-forming subunit of the sperm-specific CatSper $Ca^{2+}$ channel, which is necessary for the acquisition of an asymmetric, high-amplitude flagellar bending characteristic of hyperactivation. *CatSperd$^{Null}$* sperm are unable to cross the zona, with prior studies typically inferring this defect from the absence of fertilization (*Chung et al., 2011*; *Ren et al., 2001*).

Sperm lacking the Catsper channel are known to progressively lose motility over time (*Qi et al., 2007*) and are unable to undergo hyperactivation, resulting in failed zona penetration and infertility (*Ren et al., 2001*). Because impaired motility could potentially affect sperm binding to the zona, we first sought to quantitatively evaluate whether *CatSperd$^{Null}$* sperm were able to bind to the zona upon incubation under capacitating conditions. Thus, first, we quantified the ability of *CatSperd$^{Null}$* sperm to bind to mouse zonae in vitro or to cross the zonae in vivo. To quantify the ability of *CatSperd$^{Null}$* sperm to bind to the zona matrix, we incubated sperm from infertile *CatSperd$^{Null}$* males (no sired pups after mating for 1.5 months with fertile females) or fertile *CatSperd$^{Het}$* males (≥1 litter sired after mating with fertile females for 1.5 months) for 45 min in HTF/HSA, 37°C, 5% $CO_2$, and used either mutant sperm to separately inseminate fertile ovulated MII eggs that were previously denuded from cumulus masses by hyaluronidase (16–22 eggs, 3 replicates). We found that *CatSperd$^{Null}$* sperm bound to mouse zonae with comparable efficiency (60.2±3.55; s.e.m.) to *CatSperd$^{Het}$* sperm (66.5±1.26, s.e.m.) (*Figure 5A and B*).

While it is well known that *CatSperd$^{Null}$* sperm fail to fertilize eggs, prior studies have not quantitatively assessed their ability to reach the PVS. To address this, we used *Cd9$^{Null}$* oocytes, which enable direct measurement of sperm accumulation in the PVS and help identify defects in zona penetration. To quantify the ability of *CatSperd$^{Null}$* sperm to cross the zona, we mated five *CatSperd$^{Null}$* or *CatSperd$^{Het}$* males with five *Cd9$^{Null}$* females, whose eggs rarely fuse with sperm, leading to an accumulation of supernumerary sperm in the PVS. We found that *Cd9$^{Null}$* eggs from females mated with *CatSperd$^{Het}$* males accumulated 4.1±1.57 (s.e.m.) in the PVS, whereas there were no sperm detected in the *Cd9$^{Null}$* eggs from females mated with *CatSperd$^{Null}$* males (10 females, 10–15 eggs/female) (*Figure 5C and D*). These data confirmed the inability of *CatSperd$^{Null}$* sperm to cross the zona in vivo and are consistent with previous reports showing *CatSperd$^{Null}$* sperm being unable to fertilize eggs under in vivo conditions (*Chung et al., 2011*; *Ren et al., 2001*).

To see whether hyperactive motility is necessary for the mouse sperm to enter and cross the zebrafish micropyle, we inseminated zebrafish eggs with *CatSperd$^{Null}$*, using *CatSperd$^{Het}$* sperm as controls. Although *CatSperd$^{Null}$* sperm are unable to acquire hyperactivated motility and exhibit a decline in progressive motility over time, previous studies have shown that 20% of *CatSperd$^{Null}$* sperm retain progressive motility at 90 min (*Qi et al., 2007*). Considering that we inseminated with 100,000 progressively motile sperm, and that 15–35% of capacitated sperm are typically hyperactivated by 90 min (*Goodson et al., 2011*), we inferred that we could have relied on approximately 3000 *CatSperd$^{Null}$* sperm with progressive motility, which was indeed sufficient to locate and bind mouse zonae in vitro (*Figure 5A and B*). Therefore, we performed cross-species insemination, using *CatSperd$^{Null}$*. We rarely found *CatSperd$^{Null}$* sperm in the micropyle of zebrafish eggs and never found *CatSperd$^{Null}$* sperm in the fish ICS, indicating that CatSper is necessary for mouse sperm to enter and cross the micropyle (*Figure 5E and F*).

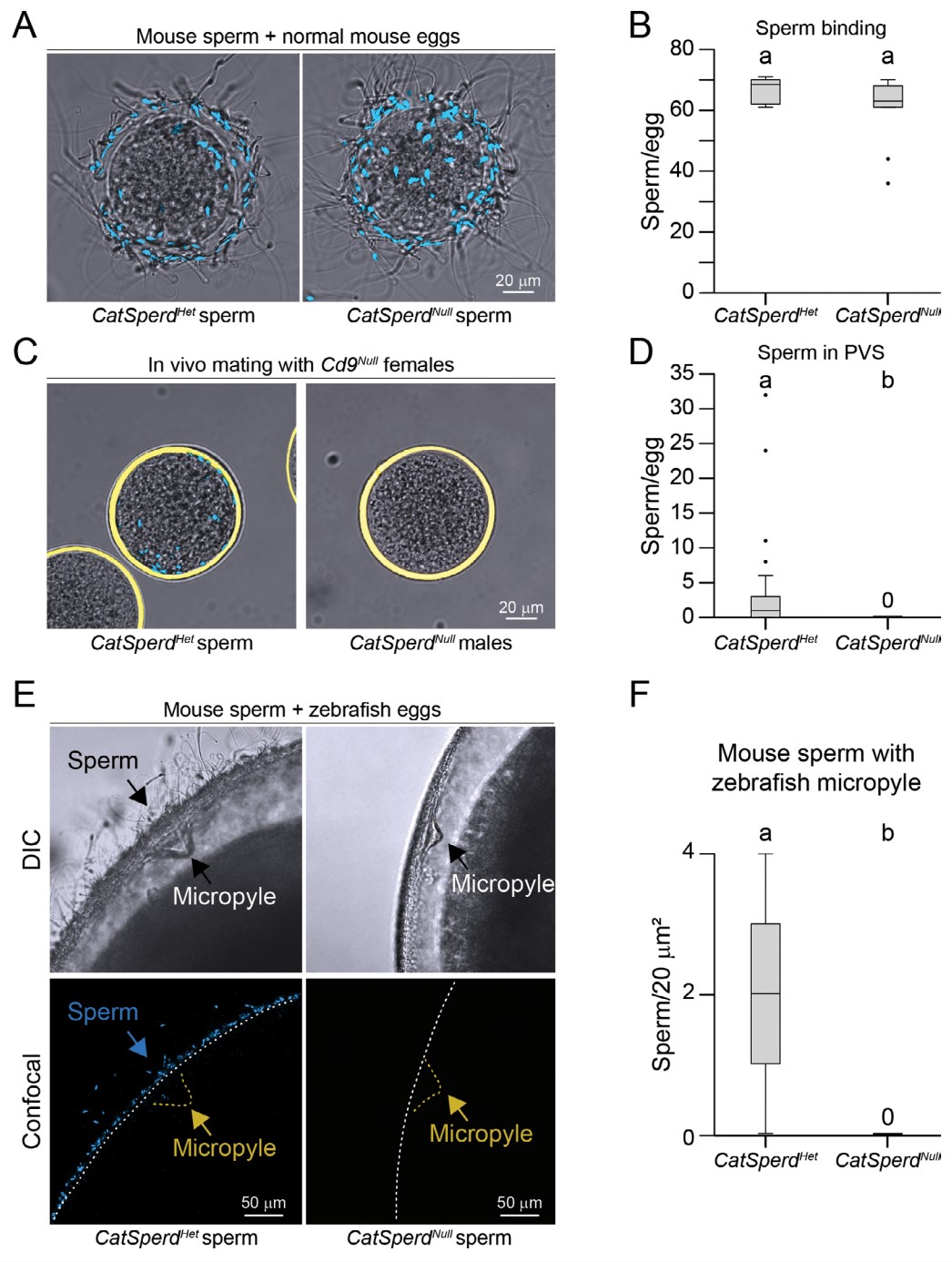

**Figure 5.** CatSper is necessary for mouse sperm crossing the zebrafish micropyle. (**A**) Hoechst-stained fertile *Catsperd^Het* (control, left) and *Catsperd^Null* sperm binding to ovulated mouse eggs. (**B**) Same as in *Figure 1D*, per mouse egg. (**C**) Fertile *Catsperd^Het* (control, left) and *Catsperd^Null* sperm in the perivitelline space (PVS) of *Cd9^Null* female eggs upon in vivo mating. (**D**) Same as in *Figure 1D*, number of PVS sperm per *Cd9^Null* mouse egg.

*Figure 5 continued on next page*

*Figure 5 continued*

(**E**) Fertile *Catsperd^Het* (control, left) and *Catsperd^Null* (right) Hoechst-stained sperm in the micropyle region of zebrafish eggs; top, differential interference contrast (DIC) image; bottom, confocal. (**F**) Same as in *Figure 1G*, with *Catsperd^Het* vs. *Catsperd^Null* sperm. B, D, and F (n=3).

## Discussion

By establishing a mouse-fish gamete interaction assay, we report that, although the chorion cannot support mouse sperm binding, a subpopulation of mouse sperm can find and enter the fish micropyle. Mouse sperm cross the micropyle and accumulate in the fish PVS. Using sperm from different transgenic mouse lines, we found that the recognition of the micropyle by mouse sperm is dependent on the CatSper channel and that the zebrafish micropyle failed to effectively induce acrosome exocytosis.

By Clustal Ω sequence alignment (*Sievers et al., 2011*), it appears that zebrafish zona proteins lack an N-terminal domain with conserved sequence identity to the mammalian ZP2 N-terminus (*Avella et al., 2014*). Therefore, the observed absence of mouse sperm binding to zebrafish ZP2 or the chorion is consistent with a model that envisions the N-terminal region of mammalian ZP2 to be required for mouse sperm binding to an extracellular matrix (*Avella et al., 2014*; *Bhakta et al., 2019*). Of note, consistent with the hypothesis that zebrafish sperm do not recognize or interact with mammalian zonae, we did not observe any zebrafish sperm bound to the mouse zona pellucida. These results highlight that cross-species insemination assays do not necessarily exhibit reciprocal sperm-egg interactions. For instance, human sperm bind specifically to human zonae but not to those of mice, whereas mouse sperm are capable of binding to both mouse and human zonae (*Avella et al., 2014*; *Baibakov et al., 2012*; *Bedford, 1977*). Such asymmetry may reflect evolutionary divergence and species-specific specialization in gamete recognition mechanisms.

A model of gamete recognition at the zona pellucida suggests that mouse sperm can cross the extracellular matrices provided they are able to bind, and this binding is regulated by the cleavage status of ZP2. In unfertilized eggs, ZP2 is uncleaved, supporting sperm binding and subsequent penetration. Following gamete fusion, cortical granule exocytosis releases the metalloprotease ovastacin, which cleaves the N-terminus of ZP2, rendering the zona unable to support additional sperm binding and thereby preventing further penetration (*Bhakta et al., 2019*). Notably, fertilized eggs or two-cell embryos that retain uncleaved ZP2 continue to support de novo sperm binding, allowing supernumerary sperm to traverse the zona and accumulate in the PVS, unable to fuse with the oolemma or blastomeres (*Baibakov et al., 2007*; *Baibakov et al., 2012*; *Burkart et al., 2012*; *Gahlay et al., 2010*). Analogously, in our experimental system, mouse sperm were observed to interact with the zebrafish micropyle opening, and this interaction was as stable as the sperm binding to the zona after washes with two-cell embryo controls. We interpret these findings to suggest that, once the interaction at the micropyle occurs, mouse sperm can cross the chorion, even under conditions in which the micropyle may have begun detaching from the oocyte due to activation. These observations imply that mouse sperm may circumvent the zebrafish oocyte's mechanism for restricting supernumerary sperm crossing the micropyle, even after egg activation has occurred.

The ability of sperm to effectively localize eggs is vital for external fertilizing species that emit their gametes in the water column. In sea urchins, small peptides such as Resact have been identified as the molecule mediating the sperm-attracting stimulus to the unfertilized egg. In zebrafish and other teleost fish species, sperm contact with water is sufficient to activate their motility, yet as they pass near the micropyle area, their motility increases (*Suzuki, 1958*). Such a gain in motility appears to be elicited by exposure to the MP that guides the fertilizing sperm into the micropyle (*Yanagimachi et al., 1992*). We and others have shown that removal of the MP by trypsin digestion significantly decreases egg fertilization rate, indicating the important role of MP in fish fertilization (*Yanagimachi et al., 2013*). Sea urchin Resact induces a cGMP signaling pathway that mediates $Ca^{2+}$ bursts via the CatSper channel, and the increased cytoplasmic $Ca^{2+}$ concentrations confer sperm chemotactic steering toward the egg (*Seifert et al., 2015*). In addition, it has been hypothesized that the fish CatSper channel may also play a role in activating fish sperm motility during fertilization (*Yanagimachi et al., 2017*). Upon binding with motile sperm, the MP may activate sperm proton pumps, leading to an increase in intracellular pH; this, in turn, would activate CatSper channels, resulting in a $Ca^{2+}$ influx and higher levels of intracellular $Ca^{2+}$. Higher concentrations of intracellular $Ca^{2+}$ would make the sperm tail beating pattern change from symmetrical to asymmetrical, directing sperm entry into the micropyle (*Yanagimachi et al., 2017*). Consistent with these hypotheses, our data demonstrates that CatSper is necessary to mediate mouse sperm entering and crossing the zebrafish micropyle. Like fish sperm, it is conceivable that the MP may elicit an attractant stimulus toward a subpopulation of mouse sperm. However, recent findings have shown successful cross-species fertilization of evolutionary-distant fish species, such as with female common carp and male blunt snout bream (*Wang et al., 2017*)

or Russian sturgeon with American paddlefish (*Káldy et al., 2020*), suggesting that such a stimulus may not act with species specificity. This is consistent with findings that show how sperm from other fish species (*Yanagimachi et al., 2013*; *Yanagimachi et al., 2017*) may reach and cross the micropyle of different fish species (*Yanagimachi et al., 2017*).

While our results show an absence of *CatSperd^{Null}* sperm accumulation within the micropyle and ICS, we acknowledge that this phenotype could be interpreted also as a consequence of general motility defects rather than a specific failure in hyperactivation or micropyle recognition. *CatSperd^{Null}* sperm are indeed characterized by an inability to develop hyperactivated motility and by a gradual decline in progressive motility over time, with only ~20% of sperm remaining motile after 90 min compared to over 70% in controls (*Qi et al., 2007*). However, under our IVF conditions, *CatSperd^{Null}* sperm maintained sufficient progressive motility within the first hour post-insemination to bind the zona with efficiency comparable to *CatSperd^{Het}* controls. Given previous reports that 15–35% of sperm achieve hyperactivation by 90 min (*Goodson et al., 2011*), and considering our use of 100,000 progressively motile sperm per insemination, we estimate that approximately 3000 hyperactivated *CatSperd^{Null}* sperm were present in the dish. Despite this, no sperm were observed in proximity to the micropyle or within the ICS. These findings support the interpretation that failure to hyperactivate, rather than total loss of motility, is the key factor preventing *CatSperd^{Null}* sperm from locating and traversing the micropyle. In addition, the inability of *CatSperd^{Null}* sperm to locate or traverse the micropyle may stem from defective rheotactic responses, which are critical for guiding sperm toward chemoattractant gradients (*Chung et al., 2017*).

The potential existence of a mammalian sperm chemoattraction mechanism has been the focus of considerable research efforts. Using in vitro assays to quantify chemoattraction in mammalian sperm (e.g. Zigmond chamber), several studies have shown evidence of a putative attraction mechanism coming from the egg microenvironment (*Eisenbach and Giojalas, 2006*), with subpopulations of sperm that responded to chemical gradients obtained from egg extracts or oviductal fluids (*Fabro et al., 2002*; *Giojalas and Rovasio, 1998*; *Oliveira et al., 1999*; *Sun et al., 2005*). However, no follow-up studies with genetic loss-of-function assays could characterize such a mechanism further. Hence, it is still uncertain whether mammalian sperm may respond to attracting stimuli from an egg (*Yanagimachi, 2022*). Our data suggests the presence of a conserved mechanism that attracts mouse sperm toward eggs. Although such stimulus comes from a fish egg, it is plausible to reason that a mammalian protein homologous to the fish MP may play a similar role toward a subpopulation of mouse sperm that swim near an ovulated egg in the ampulla. Unfortunately, studies using mass spectrometry, together with cloning, immunohistochemistry, and in situ hybridization, were unable to identify the nature of the fish MP (*Oda et al., 1995*; *Oda et al., 1998*; *Vines et al., 2002*). Therefore, this factor is still unidentified.

To successfully fertilize an egg, mammalian sperm must undergo acrosome exocytosis, as only acrosome-reacted sperm can fuse with the oolemma (*Bhakta et al., 2019*). The acrosome is a Golgi-derived organelle that underlies the anterior plasma membrane heads; it is enclosed by an inner and outer membranes and contains digestive enzymes (none found necessary for zona pellucida) (*Bhakta et al., 2019*; *Buffone et al., 2014*). During capacitation, sperm undergo the acrosome reaction, a membrane fusion event that exposes the inner acrosomal membrane and equatorial segment. Fusion with the oolemma occurs via remnants of the sperm plasma membrane overlying the equatorial segment (*Yanagimachi and Noda, 1970*). In certain fish species such as sturgeon, sperm possess an acrosome and undergo acrosome exocytosis while traversing the micropyle before gamete fusion (*Alavi et al., 2012*; *Psenicka et al., 2010*). In contrast, zebrafish sperm lack an acrosome, suggesting that the zebrafish micropyle may not have evolved mechanisms to trigger acrosome exocytosis. Consistent with this, we observed acrosome-intact mouse sperm both within the micropyle and in the ICS, indicating that passage through the zebrafish micropyle does not induce acrosome exocytosis.

In our studies, mouse sperm found in the micropyle or the ICS of zebrafish eggs were often observed to be acrosome-intact. Failed acrosome exocytosis has been observed in previous cross-species insemination studies in rodents, where sperm could penetrate zonae of different species while remaining acrosome-intact (*VandeVoort et al., 1997*; *Wakayama et al., 1996*; *Yanagimachi and Phillips, 1984*), indicating putative species-specific oocyte mechanisms of induction of acrosome exocytosis (*Avella and Dean, 2011*). As zebrafish sperm do not present an acrosome, it is conceivable that, unlike sturgeon, zebrafish oocytes may lack the ability to induce acrosome exocytosis in the micropyle.

While this study does not aim to model physiological fertilization, our findings offer insight into potential evolutionary conservation in gamete guidance mechanisms. Despite substantial evolutionary divergence between zebrafish and mice, the possibility of conserved molecular pathways guiding gamete interaction remains plausible. For example, the CatSper calcium channel, which plays a central role in mediating sperm hyperactivation, is evolutionarily conserved and functionally critical across diverse taxa, from sea urchins (external fertilizers)(*Seifert et al., 2015*) to mammals (internal fertilizers) (*Lishko and Mannowetz, 2018*). Similarly, although zebrafish sperm lack an acrosome, other fish species such as sturgeon have an acrosome that undergoes exocytosis as sperm traverse the micropyle (*Psenicka et al., 2010*). In ovoviviparous fish such as black rockfish, sperm undergo complex physiological adaptations within the female reproductive tract—including glycocalyx remodeling and immunomodulatory changes—that contribute to fertilization efficiency (*Li et al., 2024*). In mammals, capacitation encompasses the acquisition of hyperactivated motility and the competence to undergo acrosome exocytosis, both of which are essential for fertilization (*Bhakta et al., 2019*; *Puga Molina et al., 2018*; *Yanagimachi, 1957*; *Yanagimachi et al., 2017*). Notably, features functionally similar to capacitation have also been reported in external fertilizers. In the Pacific herring, sperm remain quiescent upon release into seawater and initiate motility only upon nearing the micropyle, likely in response to egg-derived signals (*Yanagimachi, 1957*; *Yanagimachi et al., 2017*). In other teleosts such as bitterling and zebrafish, sperm exhibit enhanced motility near the micropyle region (*Suzuki, 1958*; *Yanagimachi et al., 2017*). These findings collectively support the notion that sperm-egg guidance cues and changes in motility are conserved across species with distinct fertilization ecologies.

The observation that mouse sperm can locate and traverse the zebrafish micropyle, despite the absence of species-specific zona binding and the stark divergence in fertilization strategies, suggests the presence of conserved cues recognized by mammalian sperm. These results support the thesis that mechanisms regulating gamete guidance may be shared across vertebrates. The molecular identity of such attractants remains unknown, and future studies aimed at identifying these factors will be essential to better understand the mechanisms underpinning gamete attraction across species.

In addition, the existence of an MP mediating fish and mouse sperm raises the prediction for the potential existence of a mouse ortholog for the fish MP and of a cognate sperm receptor. Future studies will focus on identifying the fish MP, looking for mammalian orthologs, and characterizing its expression during fish and mouse oogenesis. In addition, mutant-null fish or mouse models will aid in characterizing the role(s) of these proteins in regulating fish and mouse fertilization.

## Materials and methods

### IACUC sentence

Experiments with zebrafish (AB line) or normal and transgenic mice were performed in compliance with the guidelines of the Animal Care and Use Committee of Sidra Medicine and the University of Tulsa under the approved animal study protocols, Sidra Medicine IACUC 2204916, and University of Tulsa TU-0050 and TU-0050R1.

### Genotyping

Transgenic mice were genotyped by PCR ([95°C for 30 s, 56°C for 30 s, 72°C for 1 min]×30 cycles, 72°C for 7 min, and 4°C for >30 min) using primers from previous publications $Acr^{Tg}$ (*Avella et al., 2016*), $CatSperd^{Null}$ (*Chung et al., 2011*; *Ren et al., 2001*), and $Cd9^{Null}$ (*Le Naour et al., 2000*).

### Light microscopy

Samples were mounted in PBS, and images of oocytes, embryos, and beads were acquired with a dissecting microscope SMZ-2B (Nikon, Japan) at ×5 magnification, with an inverted microscope ECLIPSE TS100 (Nikon) or with a confocal microscope (LSM 800, Zeiss) using 10X or 20X objective lens at room temperature (*Avella et al., 2014*). Using ZEN 3.2 image software (Zeiss), LSM 800 images were exported as 300 dpi resolution TIF files and combined using Adobe Illustrator 25.0. Alternatively, maximum intensity projections of confocal optical sections to a single plane were acquired and merged with DIC images of oocytes, embryos, or agarose beads.

### Electron microscopy

Zebrafish oocytes inseminated with mouse sperm were fixed in 2% glutaraldehyde in 0.1 M cacodylate buffer (pH 7.4) and incubated at 4°C for 2 hr. After a series of washes in cacodylate buffer, the eggs

were embedded in 2% agarose. The samples were dehydrated through a graded series of ethanol and processed for embedding in LR White resin. Ultrathin sections were obtained with an ultramicrotome (Microm International GmbH) and mounted on formvar-coated nickel grids (Electron Microscopy Sciences). Ultrathin sections were counterstained with uranyl acetate followed by lead citrate and imaged in a Jeol JEM-1011 transmission electron microscope (Jeol).

## Expression of recombinant zebrafish ZP2 and ZP3

cDNA encoding zebrafish ZP2 (AAK16577.1; 25–405 aa) and ZP3 (NP_571406; 22–396 aa) (*Wang and Gong, 1999*) were cloned into pFastBac-HBM TOPO (Invitrogen) downstream of a polyhedron promoter and a gp67 signal peptide from AcMNPV (38 aa). At the 3' end of the open reading frame, each clone carried a 6-His tag encoded in frame to enable peptide purification. Bacmid DNA, isolated after transformation into DH10Bac *Escherichia coli*, was used to transfect insect cells to produce recombinant baculovirus particles for infection of *Sf9* cells using manufacturer protocols (Genscript). Recombinant peptides were purified on IMAC Sepharose High-Performance beads (Cytiva) per the manufacturer's instructions and assayed on SDS-PAGE as previously reported (*Avella et al., 2014*). Plasmid cloning and peptide expression were performed by Genscript (USA).

## Mouse gamete collection and in vitro insemination

Female mice were stimulated with 5 IU of equine chorionic gonadotropin (ProspecBio) and 48 hr later with human chorionic gonadotropin (hCG) (Sigma-Aldrich). Twelve hours after hCG injection, eggs in cumulus were collected and incubated (37°C, 5% $CO_2$) in HTF/HSA under mineral oil before insemination with mouse sperm. Alternatively, eggs were denuded from the cumulus mass with hyaluronidase (Millipore) and washed in HTF/HSA before insemination with mouse sperm. Fertilization was scored 24 hr post-insemination by quantifying the number of two-cell embryos. To assess in vivo female fertility, mutant female mice (n≥3) were co-caged with one control fertile female and one fertile male mouse, and litters were recorded until females gave birth to at least three litters.

Epididymal sperm from 10- to 14-week-old male mice (normal or *Catsperd*[Het] and *Catsperd*[Null]) were incubated under capacitating conditions in human tubal fluid (HTF) supplemented with 0.4% human serum albumin (HSA; Cooper Surgical) for 45 min (37°C, 5% $CO_2$) (*Avella et al., 2014*). Sperm were added to oocytes or embryos in 50 µl of HTF/HSA drops under mineral oil at a final concentration of $10^5$/ml progressive motile sperm quantified by hemocytometer. Upon coincubation, inseminated oocytes/embryos were washed by gently pipetting with a 200 µl microcapillary pipette (Cooper Surgical) to remove loosely bound sperm. Samples were, then, fixed in 2% paraformaldehyde and stained with Hoechst and WGA-633 (Thermo Fisher) to identify the nuclei and the glycoproteins of the zonae pellucidae. To assess mouse sperm binding, mouse eggs and two-cell embryos were used as positive and negative controls, respectively.

## Zebrafish gamete collection and in vitro insemination

Zebrafish females or males (separated from tank mates the night prior) were anesthetized with tricaine (3-amino benzoic acid ethyl ester) solution (Sigma). After being gently dried with a kimwipe, each female was moved to a dry Petri dish, and mild pressure on the fish belly was applied, stroking from anterior to posterior. Released oocytes were separated from the female using flat forceps and inseminated with zebrafish sperm. Alternatively, zebrafish oocytes were preserved in Hank's saline (*Westerfield, 2007*) and then were gently acclimated to HTF/HSA (Cooper Surgical) and inseminated with mouse sperm previously incubated under capacitating conditions (HTF/HSA, 37°C, 5% $CO_2$) (*Avella et al., 2014*). Before insemination with mouse sperm, zebrafish oocytes were, in some instances, treated with 0.001% trypsin (crystalline, VWR) in Ringer's solution (*Westerfield, 2007*) for 1–5 min, and rinsed twice in Hank's solution before cross-species insemination (HTF/HSA, 37°C, 5% $CO_2$). Zebrafish oocytes and mouse sperm were incubated for 60 min (HTF/HSA, 37°C, 5% $CO_2$), washed in HTF/HSA to remove loosely bound sperm, and video-imaged or fixed in 2% paraformaldehyde. Zebrafish males were dried with a kimwipe, and both sides of the body were firmly stroked to collect sperm out of their genital pore with a microcapillary pipette (Cooper Surgical). Sperm were preserved in ice-cold, full-strength Hank's saline until zebrafish or mouse oocyte insemination (*Westerfield, 2007*). Upon oocyte insemination, samples were immediately fixed in 2% paraformaldehyde and stained with Hoechst and Wheat Germ Agglutinin, Alexa Fluor 633 Conjugate (WGA-633, Thermo Fisher) to identify the nuclei

and the micropyle, respectively. Fluorescence intensity of the micropyle was measured as RID using Fiji/ImageJ (*Brazill et al., 2018*).

### Live imaging of mouse sperm attraction to the micropyle of zebrafish mature eggs

Ovulated zebrafish oocytes were collected, acclimated as described above, and immediately transferred to the imaging dish containing 300 µl of HTF/BSA under mineral oil. The micropyle area was identified and brought into focus. Mouse sperm, collected and capacitated as previously described, were prepared for imaging. Images were captured using an inverted microscope (Nikon Eclipse Ti) equipped with a 12MP camera. After focusing on the micropyle, 3000 progressively motile sperm were introduced into the imaging dish, and time-lapse live imaging was performed to record gamete interactions.

### Gamete or embryo quantifications and statistics

Mouse sperm bound to oocytes or beads were quantified from z maximum intensity projections obtained by confocal microscopy (LSM800, Zeiss). The number of sperm bound per mouse oocyte/ embryo or per 20 µm$^2$ area (mouse and zebrafish oocytes/embryos) was obtained using the ZEN 3.2 software (Blue Edition, Zeiss). The micropyle region was defined as the area within a 160 µm radial distance from the micropyle opening, based on measurements obtained through confocal microscopy using ZEN Lite software (Zeiss, Germany). Sperm located within this defined region were classified as being in the micropyle area, while those beyond it were designated as 'away' from the micropyle. Descriptive statistics were used for comparisons of mouse sperm bound to oocytes, embryos, or agarose beads and for mouse sperm that have crossed the zonae with one-way ANOVA followed by Tukey's HSD (honestly significant difference) post hoc test analyses. The number of sperm was represented as bars or boxplots reflecting the median (horizontal line) and data points within the 10th and 90th percentiles (error bars) for at least three independent experiments. Boxes included the middle two quartiles, and outliers, when present, were indicated by dots. All statistical analyses were performed using R Studio (v. 3.6.3).

### AlphaFold2-multimer modeling of cross-species interactions mediating gamete membrane adhesion

To predict the protein interactions, we used 'AlphaFold2 mmseqs2' from ColabFold (*Ovchinnikov et al., 2025*; code for this article available at https://github.com/sirusb/cross-species-insemination, copy archived at *Djekidel, 2025*). The amino sequence of each protein was obtained from the UniProt data using the following IDs: mouse Spaca6 (E9Q8Q8), mouse Izumo1 (Q9D9J7), mouse Tmem81 (Q9D5K1), mouse Izumo1R (Q9EQF4), zebrafish Bouncer (P0DPQ9), zebrafish Izumo1 (A0A2R8QHQ6), zebrafish Spaca6 (A0A8M9PDM1), zebrafish Tmem81 (B8JI67). AlphaFold2 mmseqs2 was run using parameters num_relax = 1, num_recycles = 3. The PAE heatmaps and ipTM scores were used for evaluation. The distribution of the ipTM scores of the top 3 structural models between the different complexes was compared using a two-sided t-test.

## Acknowledgements

We thank Drs. Andrea Pauli and Masahito Ikawa for the critical reading of the manuscript, and Dr. Jean-Ju Chung for the *CatSperd*[Null] mice. We are grateful to Billi Bobala and Lucas Cline for their assistance during the experiments. This study was supported by the Sidra Medicine grant SDR400185; University of Tulsa, Department of Biological Science, Office of Research and Sponsored Programs (Faculty Research Grant and Faculty Research Summer Fellowship) to MA; and University of Tulsa, the Tulsa Undergraduate Research Challenge (TURC) program to ES and LG.

## Additional information

### Funding

| Funder | Grant reference number | Author |
|---|---|---|
| Sidra Medicine | SDR400185 | Matteo Avella |
| University of Tulsa | | Matteo Avella |
| University of Tulsa | TURC | Eva Stickler<br>Lillian Ghanem |

The funders had no role in study design, data collection and interpretation, or the decision to submit the work for publication.

### Author contributions

Suma Garibova, Data curation, Formal analysis, Validation, Investigation, Visualization, Methodology, Writing – original draft; Eva Stickler, Lillian Ghanem, Investigation, Methodology; Fatima AlAli, Supervision, Investigation, Visualization, Methodology; Maha A Abdulla, Investigation, Methodology, Project administration; Abbirami Sathappan, Software, Visualization, Methodology; Sahar I Da'as, Supervision; Mohamed Nadhir Djekidel, Software, Investigation; Rick Portman, Supervision, Methodology; Matteo Avella, Conceptualization, Data curation, Supervision, Funding acquisition, Investigation, Visualization, Methodology, Writing – original draft, Writing – review and editing

### Author ORCIDs

Suma Garibova ⬤ https://orcid.org/0000-0002-6651-363X
Fatima AlAli ⬤ https://orcid.org/0000-0001-6451-7357
Maha A Abdulla ⬤ https://orcid.org/0009-0008-3529-4033
Sahar I Da'as ⬤ https://orcid.org/0000-0003-3438-5028
Lillian Ghanem ⬤ https://orcid.org/0000-0001-6394-8761
Mohamed Nadhir Djekidel ⬤ https://orcid.org/0000-0001-5361-1858
Matteo Avella ⬤ https://orcid.org/0000-0003-0104-3304

### Ethics

Experiments with zebrafish (AB line) or normal and transgenic mice were performed in compliance with the guidelines of the Animal Care and Use Committee of Sidra Medicine and the University of Tulsa under the approved animal study protocols, Sidra Medicine IACUC 2204916, and University of Tulsa TU-0050 and TU-0050R1.

Reviewer #1 (Public review): https://doi.org/10.7554/eLife.106303.3.sa1
Author response https://doi.org/10.7554/eLife.106303.3.sa2

## Additional files

### Supplementary files

MDAR checklist

Source data 1. Source data for quantitative analyses, immunoblots, and microscopy measurements supporting all main and figure supplements.

### Data availability

All data generated or analysed during this study are included in the manuscript and supporting files.

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
