## [Editor Report · eLife Assessment]

This **important** study reports the conservation of sperm-egg envelope binding by demonstrating successful recognition of the micropyle in fish eggs by mouse sperm. The evidence supporting the conclusions drawn is **convincing**. This study will be of interest to reproductive biologists and clinicians studying the biology of fertilization and fertility.

---

## [Referee Report · Reviewer #1 (Public review)]

Summary:

The paper is well written and investigates the cross-species insemination of fish eggs with mouse sperm. and I have a few major and minor comments.

Strengths:

The experiments are well executed and could provide valuable insights into the complex mechanisms of fertilization in both species. I found the information presented to be very interesting,

Weaknesses:

The rationale of some of the experiments, in particular those using CatSper KO sperm is, in my view.

---

## [Author Response]

The following is the authors’ response to the original reviews.

**Reviewer #1 (Public review):**
Summary:The paper is well written and investigates the cross-species insemination of fish eggs with mouse sperm. I have a few major and minor comments.Strengths:The experiments are well executed and could provide valuable insights into the complex mechanisms of fertilization in both species. I found the information presented to be very interesting,

Thank you.

Weaknesses:The rationale of some of the experiments is not well defined.

Thank you. In the revised manuscript, we have clarified and expanded the rationale behind each experiment to better highlight the specific questions being addressed and how each approach contributes to our overall investigation. These clarifications have been integrated throughout the Results and Discussion sections. We provide detailed rationale in our point-by-point responses to both reviewers, outlining how each experimental design was motivated by prior findings, hypotheses, or specific gaps in knowledge. We hope these revisions make the experimental logic and progression better defined and more compelling.

Major Comments:(1) Figure 5I do not understand the rationale for performing experiments using CatSper-null sperm and CD9-null oocytes. It is well established that CatSper-null sperm are unable to penetrate the zona pellucida (ZP), so the relevance of this approach is unclear.

We thank the reviewer for this comment. This experiment was conducted as the basis to then evaluate the contributions of progressive and hyperactivated motility to the ability of mouse sperm to locate and traverse the zebrafish micropyle. In earlier experiments (Figures 1 and 3), we assessed whether sperm-micropyle interaction was robust by comparing it to binding to the mouse zona pellucida and testing whether both interactions persisted after washing, which is standard approach to distinguish specific binding from non-specific adherence (Avella et al., 2014; Baibakov et al., 2012). Thus, we extended this analysis to CatSper1^Null^ sperm; CatSper1^Null^ sperm were still capable of binding the zona pellucida comparably to heterozygous controls, though they were unable to cross the zona of Cd9^Null^ eggs. These observations served as a validation step for the use of CatSper1^Null^ sperm for downstream micropyle interaction assays. Thus, we proceeded to test whether hyperactivated motility, absent in CatSper1^Null^ sperm, is required for locating and crossing the micropyle.

It is indeed well established that CatSper1^Null^ sperm are unable to penetrate the zona pellucida, and previous studies have typically used the absence of fertilized eggs as a readout. However, failed fertilization may result from multiple factors, including impaired sperm motility, reduced capacity to bind the zona pellucida, or an inability to penetrate it. To our knowledge, no study has quantitatively assessed the number of CatSper-deficient sperm that successfully bind, cross the zona and reach the perivitelline space. To address this, we first used normal oocytes for sperm binding and Cd9^Null^ oocytes (Le Naour et al., 2000), which allow direct quantification of sperm accumulation in the perivitelline space. We have 7included a detailed explanation in the Results to clarify this point, lines 352-365 and 376-369.

(2) Micropyle penetration and sperm motilityCatSper-null sperm are reportedly unable to cross the micropyle, but this could be due to their reduced motility rather than a lack of hyperactivation per se. Were these experiments conducted using capacitated or non-capacitated spermatozoa? What was the observed motility of CatSper-null sperm during these assays? Clarifying these conditions is essential to avoid drawing incorrect conclusions from the results.

Thank you for raising these points. Under our IVF conditions, qualitative observations confirmed that CatSper1^Null^ sperm displayed progressive motility, maintained sufficient progressive motility during the first hour post-insemination and exhibited zona binding efficiency comparable to that of CatSper1^Het^ controls (Figure 5A and B). This is consistent with previous reports showing that within the first 90 minutes of sperm incubation in media, approximately 20% of CatSper1^Null^ sperm preserve motility (Qi et al., 2007). Given previous studies indicating that 15–35% of sperm undergo hyperactivation within 90 minutes (Goodson et al., 2011), and considering that 100,000 progressively motile sperm were used for insemination, we estimate that approximately 3,000 hyperactivated CatSper1^Null^ sperm were present in the cross-species insemination dish (mouse sperm x zebrafish eggs). Based on these numbers, we would have expected at least some sperm to locate the micropyle if hyperactivation were not required for its detection and entry. Nevertheless, CatSper1^Null^ sperm were detected in proximity to the micropyle canal, its opening, or within the inter-chorion space (ICS). These observations support the conclusion that the inability ofCatSper1^Null^ sperm to locate and enter the micropyle is attributable to their failure to hyperactivate. Also, all sperm used in these assays were exposed to identical capacitating conditions (HTF/HSA, 37 °C, 5% CO2). We now clarify this in the Methods, line 624, and we added more rationale under the Results, lines 361-365 and in the Discussion, lines 470-483.

(3) Rheotaxis and micropyle navigationPrevious studies have shown that CatSper-null sperm fail to undergo rheotaxis. Could this defect be related to their inability to locate and penetrate the micropyle? Exploring a potential shared mechanism could be informative.

Thank you for raising this interesting point. Indeed, homozygous mutant mice lacking expression of a different component of the CatSper channel, CatSperz, show reduced rheotactic efficiency and severe subfertility (Chung et al., 2017). We cannot exclude that complete lack of CatSper as shown in CatSper1^Null^ mice could lead to reduced rheotactic efficiency, hence we include this interpretation in the Discussion (lines 484-486).

(4) Lines 61-74This paragraph omits important information regarding acrosomal exocytosis, which occurs prior to sperm-egg fusion. Including this detail would strengthen the discussion.

Thank you. We have revised the text in the discussion to describe the process of acrosome exocytosis, and its relevance for fertilization (lines 504-518).

**Reviewer #2 (Public review):**
Summary:Garibova et al. investigated the conservation of sperm recognition and interaction with the egg envelope in two groups of distantly related animals: mammals (mouse) and fish (zebrafish). Previous work and key physiological differences between these two animal groups strongly suggest that mouse sperm would be incapable of interaction with the zebrafish egg envelope (chorion) and its constituent proteins, though homologous to the mammalian zona pellucida (ZP). Indeed, the authors showed that mouse sperm do not bind recombinant zebrafish ZP proteins nor the intact chorion. Surprisingly, however, mouse sperm are able to locate and bind to the zebrafish micropyle, a specialized canal within the chorion that serves as the egg's entry point for sperm. This study suggests that sperm attraction to the egg might be highly conserved from fish to mammals and depends on the presence of a still unknown glycosylated protein within the micropyle. The authors further demonstrate that mouse sperm are able to enter the micropyle and accumulate within the intrachorionic space, potentially through a CatSper-dependent mechanism.Strengths:The authors convincingly demonstrate that mouse sperm do not bind zebrafish ZP proteins or the chorion. Furthermore, they make the interesting observation that mouse sperm are able to locate and enter the zebrafish micropyle in an MP-dependent manner, which is quite unexpected given the large evolutionary distance between these species, the many physiological differences between mouse and zebrafish gametes, and the largely different modes of both fertilization and reproduction in these species. This may indicate that the sperm chemoattractant in the egg is conserved between mammals and fish; however, whether zebrafish sperm are attracted to mouse eggs was not tested.

Thank you. We performed an additional experiment with fish sperm used to inseminate ovulated mouse eggs, and results are reported in lines 183-187 and in Supplementary Figure 2.

Weaknesses:The key weakness of this study lies in the rationale behind the overall investigation. In mammals, the zona pellucida (ZP) has been implicated in binding sperm in a taxon-specific manner, such that human sperm are incapable of binding the mouse ZP. Indeed, work by the corresponding author showed that this specificity is mediated by the N-terminal region of the ZP protein ZP2 (Avella et al., 2014). The N-termini of human and mouse ZP2 share 48% identity, which is higher than the overall identity between mouse and zebrafish ZP2, with the latter ortholog entirely lacking the N-terminal domain that is essential for sperm binding to the ZP. Given this known specificity for mouse vs. human sperm-ZP binding, it does not follow that mouse sperm would bind ZP proteins from not only a species that is much more distantly related, but also one that is not even a mammal, the zebrafish. Furthermore, the fish chorion does not play a role in sperm binding at all, while the mammalian ZP can bind sperm at any location. On the contrary, the zebrafish chorion prevents polyspermy by limiting sperm entry to the single micropyle.

We thank the reviewer for this detailed comment. In this study, our goal was precisely that one of validating the hypothesis that mouse sperm would not bind either recombinant fish ZP proteins or the chorion; in addition, we found it important to examine the observation that mouse sperm could detect the micropyle. We further elaborated this rationale in the Introduction (lines 93-100).

In addition, though able to provide some information regarding the broad conservation of sperm-egg interaction mechanisms, the biological relevance of these findings is difficult to describe. Fish and mammals are not only two very distinct and distantly related animal groups but also employ opposite modes of fertilization and reproduction (external vs. internal, oviparous vs viviparous). Fish gametes interact in a very different environment compared to mammals and lack many typically mammalian features of fertilization (e.g., sperm capacitation, presence of an acrosome, interaction with the female reproductive tract), making it difficult to make any physiologically relevant claims from this study. While this study may indicate conserved mechanisms of sperm attraction to the egg, the identity of the molecular players involved is not investigated. With this knowledge, the reader is forced to question the motivation behind much of the study.

We thank the reviewer for their perspective, and we appreciate the opportunity to further elaborate on our rationale. As outlined in our Results and Discussion sections, a growing body of evidence supports the presence of conserved molecular players and signaling pathways involved in gamete interaction across species with diverse reproductive strategies. While zebrafish and mice do differ in their fertilization environments and modes of reproduction, these differences may not necessarily exclude the possibility of conserved molecular mechanisms underlying gamete interaction. For example, the CatSper calcium channel, which plays a key role in regulating sperm motility and hyperactivation, is conserved across a broad range of taxa—from echinoderms such as sea urchins (external fertilizers)(Seifert et al., 2015) to mammals, including mice and humans (internal fertilizers)(Lishko and Mannowetz, 2018). Moreover, sperm from some fish species possess acrosomes that undergo exocytosis prior to fertilization while sperm cross the micropyle (Psenicka et al., 2010). Also, in ovoviviparous species with internal fertilization, such as the black rockfish, sperm do undergo molecular changes while in the female reproductive tract—including immunomodulatory adaptations, glycocalyx remodeling, and interactions with ovarian cells—enabling the sperm with a longer-term survival and a selective persistence that ensures only the fittest sperm can successfully fertilize eggs (Li et al., 2024). As per the mammalian capacitation, it is broadly defined as the process during which sperm undergo hyperactivation (Yanagimachi, 1970), and acquire the ability to undergo the acrosome exocytosis, making the sperm competent for gamete fusion and fertilization (Bhakta et al., 2019; Puga Molina et al., 2018; Yanagimachi, 1957; Yanagimachi et al., 2017). Of note, acrosome exocytosis or changes in sperm motility are not exclusive to internal fertilizers. For example, as we cite in our manuscript (and as just stated above), acrosome exocytosis has been described to occur as sturgeon sperm cross the micropyle (Psenicka et al., 2010). As per changes in flagellar motility, investigations in the Pacific herring (Clupea sp.) demonstrated that sperm remain nearly immotile upon release into seawater and only initiate motility when approaching the micropyle region of the egg (Yanagimachi, 1957; Yanagimachi et al., 2017). In other fish, including bitterling and zebrafish, further enhancement in sperm motility is observed as sperm approach the micropyle area (Suzuki, 1958; Yanagimachi et al., 2017). These studies suggest that functional equivalents of capacitation may exist across taxa.

We interpret the observation that mouse sperm can locate and enter the micropyle as suggesting that underlying guidance mechanisms may be more broadly conserved across distant species than previously recognized. We have now elaborated on these points in the revised Discussion (lines 531-552), and we hope the motivation behind our study is now more clearly articulated.

During fertilization in fish, the sperm enters the micropyle and subsequently, the egg, as it is simultaneously activated by exposure to water. During egg activation, the chorion lifts as it separates from the egg and fills with water. This mechanism prevents supernumerary sperm from entering the egg after the successfully fertilizing sperm has bound and fused. In this study, the authors show that mouse sperm enter the micropyle and accumulate in the intrachorionic space. Whether any sperm successfully entered the egg is not addressed, and the status of egg activation is not reported.

We appreciate the reviewer’s detailed comments and the opportunity to elaborate on this important aspect for our cross-insemination assay. We interpret the reviewer’s reference to “sperm entering the egg” as pertaining to sperm adhesion to the oocyte plasma membrane followed by fusion with the egg cell, two separate steps regulated by different molecular players for sperm-egg plasma membrane adhesion (Bianchi et al., 2014; Fujihara et al., 2021; Herberg et al., 2018; Inoue et al., 2005) and for fusion. It is important to note that proteins mediating gamete fusion are still unidentified in fish and mammals (Bianchi and Wright, 2020; Deneke and Pauli, 2021).

In our cross-species insemination experiments, zebrafish oocytes were maintained in Hank’s solution to limit spontaneous activation; however, as the reviewer correctly notes, activation likely occurred upon exposure to HTF. While this model does not recapitulate full fertilization events, it serves as a platform to explore whether mammalian sperm can detect (within the scope of our study) and respond (future studies) to putative evolutionarily conserved signals, such as those guiding fish sperm toward the micropyle.

While investigating cross-species sperm–oocyte fusion was not within the scope of this study and would require a distinct set of experimental approaches, we believe this question is an important one. However, we do not expect our platform to be informative for evaluating sperm adhesion to the fish oolemma or for enabling cross-species gamete fusion. In our assays focused on sperm-micropyle interaction, Hoechst staining of nuclei of transgenically-tagged acrosome sperm revealed no evidence of sperm adhesion to or fusion with the fish egg membrane (Figure 4D). Also, molecular incompatibilities may further prevent this interaction: in zebrafish, the Ly6/uPAR family protein Bouncer is expressed exclusively in the egg and is necessary for sperm–egg membrane adhesion (Herberg et al., 2018). Recent studies in zebrafish and mice have shown that a conserved trimeric complex composed of Izumo1, Spaca6, and Tmem81 on the sperm surface is required for mediating adhesion to the oocyte membrane by interacting with the mammalian oocyte receptor Izumo1R (also known as JUNO) or the zebrafish oocyte receptor Bouncer (Deneke et al., 2024). One would hypothesize that for mouse sperm to adhere to the zebrafish egg membrane, the mouse Izumo1-Spaca6-Tmem81 complex would need to establish binding with Bouncer. To explore this possibility, we performed AlphaFold2-Multimer structural predictions and docking analyses to mimic an interaction between mouse Izumo1-Spaca6-Tmem81 and zebrafish Bouncer, using mouse Izumo1-Spaca6-Tmem81 and Juno or zebrafish Izumo1-Spaca6-Tmem81 and Bouncer as positive controls. We observed low binding affinity between zebrafish Bouncer and the mouse trimeric complex (Izumo1, Spaca6, and Tmem81), as indicated by low ipTM scores and high predicted aligned error (PAE) values. These findings suggest that the mouse complex is unlikely to form an interaction with Bouncer (now shown in Suppl. Figure 7). These predictions were consistent with our observations that no sperm were found adhering or fusing to the egg cell. We describe methods and results in the supplementary files (Supporting Info, lines 53-66) and in the result sections (lines 335-339).

In Supplementary Videos 3-4, the egg shown has been activated for some time, as evident by the separation of yolk and cytoplasm, yet the chorion is only partially expanded (likely due to mouse IVF conditions). How multiple sperm were able to enter the micropyle but presumably not the egg is not addressed, yet this suggests that the zebrafish mechanism of blocking polyspermy (fertilization by multiple sperm) is not effective for mouse sperm or is rendered ineffective due to mouse IVF conditions. The authors do not discuss these observations in the context of either species' physiological process of fertilization, highlighting the lack of biological context in interpreting the results.

Thank you for raising this important point. One model for mammalian gamete recognition at the zona supports the notion that mouse sperm can penetrate extracellular matrices as long as sperm can bind to them, and binding is dependent on the cleavage status of ZP2. Zonae surrounding unfertilized mouse eggs present uncleaved ZP2 and these zonae support sperm binding. After gamete fusion, the cortical granules release ovastacin which cleaves ZP2 at the N-terminus, and consequently, zonae presenting cleaved ZP2 no longer support sperm binding. This mechanism acts as block to zona binding and prevents further crossing (Bhakta et al., 2019). Indeed, fertilized mouse eggs or 2-cell embryos surrounded by a zona containing uncleaved ZP2 support de novo sperm binding, and supernumerary sperm cross the zona and accumulate in the perivitelline space, unable to fuse with the fertilized oocyte plasma membrane or blastomere cells (Baibakov et al., 2012, 2007; Burkart et al., 2012; Gahlay et al., 2010). Thus, because under our experimental conditions, mouse sperm could interact with the micropyle opening, we interpret these findings to suggest that once interaction occurs at the micropyle opening, mouse sperm are capable of crossing it, even under conditions where the micropyle may be detached from the oocyte due to oocyte activation. Therefore, our data indicates that mouse sperm may be able to bypass the mechanism of zebrafish oocytes blocking multiple sperm to pass through the micropyle, even after oocyte activation. This point has now been incorporated into the revised Discussion (lines 425-441).

The authors further show that the zebrafish micropyle does not trigger the acrosome reaction in mouse sperm. Whether the acrosome reacts is not correlated with a sperm's ability to cross the micropyle opening, as both acrosome-intact and acrosome-reacted sperm were observed within the intrachorionic space. While the acrosome reaction is a key event during mammalian fertilization and is required for sperm to fertilize the egg, zebrafish sperm do not contain an acrosome. Thus, these results are particularly difficult to interpret biologically, bringing into question whether this observation has biological relevance or is a byproduct of egg activation/chorion lifting that indirectly draws sperm into the chorion.

We thank the reviewer for raising this point and we appreciate the opportunity to elaborate on the biological relevance of this experiment. Our motivation to assess acrosome status in mouse sperm following entry into the zebrafish micropyle stemmed from the following biological considerations. In fish species such as the sturgeon, sperm present an acrosome and undergo acrosome exocytosis while passing through the micropyle, before gamete fusion (Alavi et al., 2012; Psenicka et al., 2010). By contrast, zebrafish sperm lack an acrosome, raising the hypothesis that the zebrafish micropyle may not be able to trigger acrosome exocytosis. However, this possibility has not been experimentally tested. We therefore considered it important to investigate whether passage through the zebrafish micropyle induces acrosome exocytosis in mouse sperm. We have revised the Discussion to better clarify the rationale behind the experiment as well as the interpretation of the findings (lines 504-518). As per the chorion lifting indirectly drawing sperm into the chorion, we have not observed this phenomenon.

The final experiments regarding CatSper1's role in mediating mouse sperm entry into the micropyle/chorion are not convincing. As no molecular interactions are described or perturbed, the reader cannot be sure whether the sperm's failure to enter is due to signaling via CatSper1 or whether the overall failure to undergo hyperactivation limits sperm motility such that the mutant sperm can no longer find and enter the zebrafish micropyle. Indeed, in Figure 5E, no CatSper1 mutant sperm are visible near any part of the egg, suggesting that overall motility is impaired, and this is not a phenotype specific to interactions with the micropyle.

We appreciate the comment and the opportunity to further elaborate on the rationale of this experiment. While our data demonstrates a lack ofCatSper1^Null^ sperm accumulation within the micropyle and ICS, we appreciate that this may be interpreted as the result of general motility defects, rather than a specific failure in undergoing hyperactivation and micropyle recognition. CatSper1^Null^ sperm are known to lack hyperactivated motility and exhibit a progressive loss of forward motility over time. After 90 minutes, only ~20% of CatSper1^Null^l sperm remain motile, compared to over 70% in fertile sperm (Qi et al., 2007). Of note, under our IVF conditions, CatSper1^Null^ sperm retained sufficient progressive motility during the first hour post-insemination to bind the zona pellucida with comparable efficiency to CatSper1^Het^ controls. Based on prior reports indicating that 15–35% of sperm exhibit hyperactivation by 90 minutes (Goodson et al., 2011), and considering that we inseminated with 100,000 progressively motile sperm, we estimate that approximately 3,000 hyperactivated CatSper1^Null^ sperm were present in the dish. Yet, none were observed near the micropyle canal, its opening, or within the ICS. This led us to conclude that failure to hyperactivate underlies the inability of CatSper1^Null^ sperm to reach and traverse the micropyle. Also, we appreciate that identifying the molecular components of the micropyle would allow direct testing of whether the CatSper channel is activated in response to micropyle-associated signals. Indeed, no targeted perturbation of molecular interaction regulating micropyle recognition was performed in this study, as the molecular identity of the zebrafish micropyle guidance cue remains unknown. Efforts to identify and characterize this factor are ongoing in our lab and lie outside the scope of the current work. Therefore, throughout the manuscript, we have clarified that it is the failure to undergo hyperactivation, rather than the absence of CatSper per se, that limits the ability of sperm to locate and traverse the micropyle. The rationale for the experiment, the interpretation of our findings, and relevant future directions have been further elaborated in the revised Abstract, Impact Statement and Discussion (lines 40-41; 46-47; 343-365; 376-379; 389-399; 470-486).

**Reviewer #1 (Recommendations for the authors):**
Minor Comments(1) Figure numberingThere appear to be inconsistencies in the figure references. For example, what is referred to as Figure 3F in the text is actually Figure 4F. Please review and correct all figure labels for accuracy.

We thank the reviewer for pointing this out. We have carefully reviewed the manuscript and corrected all figure references throughout the text. Also, for better flow and coherence, we have moved the paragraph describing the videos to the end of the Results section titled "Mouse sperm recognize the micropylar region of fish oocytes." Previously, the callout of panels in Figure 3 was out of order (3A, 3B, 3E, 3C, 3D), and this reorganization also helps maintain logical progression through the figure panels.

(2) Figure 5 terminology:The term "normal" sperm should be replaced with "CatSper heterozygous (Het)" sperm to avoid confusion and improve precision.

We thank the reviewer for this helpful suggestion. We have revised the terminology in Figure 5 and throughout the manuscript, replacing “normal” sperm with “CatSper1 heterozygous (Het)”

**Reviewer #2 (Recommendations for the authors):**
In addition to my comments in the public review, I would encourage the authors to consider the following suggestions:The authors show that mouse sperm can find and enter the fish micropyle, and that this depends on the presence of MP. To better assess sperm binding to the micropyle region, the number of sperm binding to the micropyle vs. non-micropyle chorion should be clearly quantified, as well as the percentage of sperm that enter the micropyle compared to the total used for insemination. The authors state several times throughout the text that a "subpopulation" of mouse sperm finds and enters the micropyle, but it would be more precise and informative to give a percentage.

We thank the reviewer for this suggestion. We have now reported also the number of sperm bound to the other regions of the chorion (away; lines 231-233), as well as the percentage of sperm that entered the micropyle relative to the total number used for insemination (lines 276-279).

To ensure that all sperm are inside the chorion, the egg should be removed from the insemination dish, washed thoroughly, and then the chorion should be torn open to definitively show that the sperm were indeed inside.

We thank the reviewer for these excellent suggestions. As per ensuring that the sperm are inside the ICS, (as shown now in Figures 4A, F, G , Supplementary Figure 6 and Supplementary Movies 3–5), the inseminated oocytes were thoroughly washed prior to imaging to ensure that only sperm located inside the chorion were visualized (as described in the Methods, lines 646-648). In addition, to confirm the spatial localization of sperm within the ICS, we are now including additional TEM images showing sperm in the ICS (Figure 4G, right panel). Also, we generated orthogonal views using ZEN Lite software (Zeiss, Germany) from a z-stack encompassing the full volume of the chorion, ICS, and oocyte (added in the supplementary materials, as Supplementary Figure 6). These views display three focal planes: the surface of the WGA-stained chorion, the middle of the ICS, and the oocyte plasma membrane. Sperm nuclei stained with Hoechst are clearly visible below the chorion surface and above the oocyte plasma membrane, confirming their localization within the ICS. Additionally, in a separate set of experiments, as recommended by this reviewer, we mechanically disrupted the chorion and consistently detected sperm within the ICS. This procedure, however, was technically challenging: upon disruption, the chorion often collapsed onto the oocyte, and during the extraction process, sperm were sometimes displaced. As a result, it was not always possible to determine with complete confidence whether the sperm had originally been located inside or outside the chorion. However, we hope that the additional TEM and confocal images (Figure 4G and Supplementary Figure 6) offer further support for the localization of sperm within the ICS.

I would further suggest that they examine the micropyle opening after the entry of multiple sperm, as well as the dynamics of egg activation during insemination with mouse sperm.

Thank you. We now include one additional TEM image capturing the full structure of a micropyle that was traversed by multiple mouse sperm (shown in Figure 4G, left panel).

At what point does the micropyle detach from the egg surface? Live imaging of this process with a confocal microscope would be very informative.

During live imaging, the interval between placing the oocyte in the imaging dish, replacement of Hank’s solution with HTF and the addition of sperm, followed by the initiation of video acquisition, is approximately 2 to 3 min. By this time, the ICS is already apparent (Supplementary Video 2), although the micropyle appears to remain adherent to the egg cell. Partial detachment of the micropyle from the egg cell begins around 6–7 minutes after imaging starts and continues progressively over time. We provide time-lapse imaging frames to show the micropyle detachment under mouse IVF conditions (Supplementary Figure 5).

Along the same lines, sperm should be doubly labeled with an acrosome-independent marker, i.e., a live DNA stain or MitoTracker. Then the authors could track if any sperm are actually able to enter the egg itself, which would be highly unlikely but an important detail to confirm.

Thank you for pointing this out. In our assays designed to study sperm–micropyle interactions, Hoechst staining of nuclei in transgenically labeled acrosome sperm showed no indication of sperm adhesion to, or fusion with, the zebrafish egg cell (Figure 4D).

Line 242, 282: The text should refer to Figure 4, not 3. Please make sure all figure references correspond to the correct figure and panel.

Thank you for bringing this to our attention. We have carefully reviewed the manuscript and corrected the reference to Figure 4, along with all other figure and panel citations to ensure they accurately correspond to the correct content. Also, to improve the overall flow, we relocated the paragraph describing the videos to the end of the Results section titled "Mouse sperm recognize the micropylar region of fish oocytes". This change also helped correct the sequence of figure panel references, which were previously cited out of order (i.e., 3A, 3B, 3E, 3C, 3D).

Line 244: The authors quantify sperm that are "away" from the micropyle, but this is not clearly defined. This should be given as a set radius or distance from the center (e.g., in microns). If the sperm are still motile, can this be accurately measured?

We thank the reviewer for this valuable suggestion. We have now defined “away from the micropyle” as a distance greater than 160 µm from the center of the micropyle. This measurement was determined using confocal z-stack projections of fixed samples. These details have been added to the revised Methods section (lines 670-674).

To strengthen the conclusion that the sperm chemoattractant is indeed conserved from fish to mammals, the authors could show that zebrafish sperm are also able to find/approach mouse eggs. Even more compelling would be to show the same is true for other species combinations. As it stands, the choice of comparing mouse and zebrafish does not seem scientifically motivated but rather due to their availability.

We thank the reviewer for this important suggestion. To test whether zebrafish sperm are capable of binding to the mammalian zona pellucida, we conducted the suggested experiment: ovulated, cumulus-free mouse oocytes were placed in water and incubated with zebrafish sperm. We did not observe any zebrafish sperm bound to the mouse zona pellucida, consistent with the hypothesis that zebrafish sperm do not recognize or interact with mammalian zonae or ZP proteins. This has now been added in the Results (lines 183-187) and shown in Supplementary Figure 2. We interpret these findings as in cross-species insemination assays, reciprocity in sperm-egg interaction is not always observed. For example, while human sperm bind only to human zonae and not to mouse zonae, mouse sperm are able to bind both mouse and human zonae (Avella et al., 2014; Baibakov et al., 2012; Bedford, 1977). This asymmetry may reflect species-specific adaptations in sperm-egg recognition. We have now added this point to the revised Discussion to clarify the rationale and context of our approach (lines 416-423).

As per the choice of experimental models, while we agree that testing additional species combinations would broaden the scope of the findings, the choice to compare mouse and zebrafish was not solely based on availability. Rather, it was motivated by the opportunity to examine sperm guidance across two evolutionary distant vertebrates. This contrast allows us to seek for potential conservation of structural or molecular cues involved in gamete interaction. Additionally, both zebrafish and mouse offer extensive gene editing, blotting and imaging reagents, which are particularly valuable should future studies aim to identify and functionally disrupt genes encoding micropyle-associated proteins and their putative orthologs in mammals.

For the CatSper experiment, I would suggest that the authors repeat this experiment with another mouse sperm mutant that is known to have reduced/altered motility. With the current data, I do not believe the failure to find/enter the micropyle is necessarily CatSper-specific. Because we do not know what the sperm interacts with in the micropyle or what the MP interacts with on the sperm, the signaling pathway cannot be tested, making other controls necessary for these results to be meaningful.

Thank you for highlighting this important point. A wide range of mouse models with sperm motility defects exhibit subfertility or infertility due to structural abnormalities in the axoneme or midpiece rigidity. (Miyata et al., 2024). These defects often result in impaired progressive motility, failure to reach the zona pellucida, or inability to bind or penetrate it. In contrast, we could test and validate that CatSper1^Null^ sperm display preserved early progressive motility but fail to transition into hyperactivated motility, making them particularly well suited for specifically assessing the role of hyperactivation in sperm navigation toward and entry into the micropyle. Taken together, these points, along with those discussed in our response to the public review, led us to conclude that the CatSper1^Null^ model provides the most biologically relevant context currently available to assess the role of hyperactivation in guiding sperm to the micropyle.

The authors could greatly strengthen the discussion by addressing the key points I raised in the public review, particularly in terms of interpreting these results in the context of each species' physiological mode of fertilization.

We thank the reviewer for this important recommendation. We have carefully revised the Discussion to address the key points raised in the public review, particularly by framing our findings within the context of the distinct physiological modes of fertilization in each species, as indicated n our answers to the public review. We hope these additions have strengthened the manuscript as suggested.